# Measuring, Evaluating and Improving Logical Consistency in Large Language Models

## Abstract

Recent research in Large Language Models (LLMs) has shown promising progress related to LLM alignment with human preferences. LLM-empowered decision-making systems are expected to be predictable, reliable and trustworthy, which implies being free from paradoxes or contradictions that could undermine their credibility and validity. However, LLMs still exhibit inconsistent and biased behaviour when making decisions or judgements. In this work, we focus on studying *logical consistency of LLMs* as a prerequisite for more reliable and trustworthy systems. Logical consistency ensures that decisions are based on a stable and coherent understanding of the problem, reducing the risk of erratic or contradictory outputs. We first propose a universal framework to quantify the logical consistency via three fundamental proxies: *transitivity*, *commutativity* and *negation invariance*. We then evaluate logical consistency, using the defined measures, of a wide range of LLMs, demonstrating that it can serve as a strong proxy for overall robustness. Additionally, we introduce a data refinement and augmentation technique that enhances the logical consistency of LLMs without sacrificing alignment to human preferences. It augments noisy and sparse pairwise-comparison annotations by estimating a partially or totally ordered preference rankings using rank aggregation methods. Finally, we show that logical consistency impacts the performance of LLM-based logic-dependent algorithms, where LLMs serve as logical operators.

## 1 Introduction

Recent research in Large Language Models (LLMs; Brown et al. 2020; OpenAI 2023; Anil et al. 2023a;b) has achieved substantial progress concerning their instruction-following capabilities and generating responses aligned with human preferences. Consequently, these advancements have broadened their application to an even wider range of complex natural language tasks across diverse domains, often with minimum or even no supervised data (Kojima et al., 2022). The instruction-following abilities of LLMs are primarily achieved through supervised training with instruction-following data (Wei et al., 2022a; Chung et al., 2022) and reinforcement learning from human feedback (RLHF; Christiano et al. 2017; Stiennon et al. 2020; Ouyang et al. 2022b). During the alignment phases, LLMs are enhanced to align with human values, enabling them to generate responses that better support their decision-making and problem-solving (Dai et al., 2024).

Despite these advancements, key challenges still exist regarding the reliability and trustworthiness of LLMs. Issues such as hallucination (Zhang et al., 2023), bias (Gallegos et al., 2024), and inconsistencies in reasoning (Huang & Chang, 2023) continue to affect their credibility. These limitations hinder the full practical deployment of LLMs, particularly in professional and high-stakes applications where *trustworthiness* and *reliability* are crucial.[1] The foundation of a reliable and trustworthy system is the **consistency** of its predictions. A consistent system produces explainable and tractable decisions, enhancing its dependability and reliability. In this work, we focus on a key form of consistency in LLMs: **logical consistency**. This is especially critical for applications requiring structured reasoning and coherent decision-making (Creswell et al., 2023). Logical inconsistencies can lead to unreliable conclusions (Restall, 2002) and even paradoxes (Hyde, 2011), posing significant

---

[1]For instance, in the fields of behavioral economics and psychology systems are traditionally evaluated on two key dimensions: *validity* and *reliability* (Schmidt et al., 2000; Guion, 2004; Miller et al., 2023).

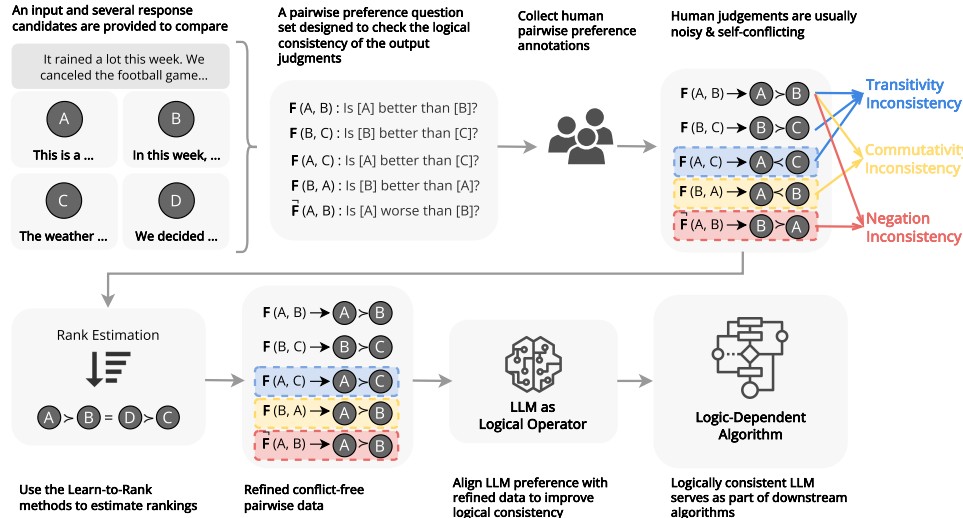

Figure 1: Three types of logical inconsistencies are observed in real-world pairwise annotations (top row): Transitivity, Commutativity, and Negation Invariance. By refining the data for self-consistency using rank estimation, we can train LLMs with enhanced logical consistency, improving their performance in logic-dependent algorithms (bottom row).

risks in domains that demand rigorous logical judgment, such as temporal or spatial reasoning (Mostafazadeh et al., 2016b), optimization (Guo et al., 2024), and automated decision systems.

In this paper, we focus on three fundamental aspects of logical consistency: **transitivity**, **commutativity**, and **negation invariance**. These properties are crucial for a coherent understanding of relationships between items, and our goal is to measure, evaluate, and enhance them in LLMs. We propose a universal framework for quantifying these consistency properties, applicable to any number of items, supporting both partial and complete comparisons, and adaptable across various domains. Our empirical investigation spans a diverse set of LLMs, including Llama 3 (Dubey et al., 2024), Gemma (Rivière et al., 2024), and Phi-3 (Abdin et al., 2024), among others. Through extensive evaluations across three representative domains, we find a strong correlation between logical consistency and the internal robustness and reliability of LLMs. This suggests that logical consistency could be a valuable (proxy) indicator of overall model reliability.

To enhance the logical consistency of LLM judgements, we propose a data refinement and augmentation framework that improves noisy and sparse pairwise comparison annotations by estimating preference rankings using rank aggregation methods. The framework then augments the data by 'logically extrapolating' additional pairwise comparisons from these rankings. Our results show that LLMs trained with this approach achieve improved internal logical consistency without compromising alignment with human preferences. Furthermore, we demonstrate that, when applied to logic-dependent downstream applications, LLMs with better logical consistency outperform less consistent models. They show better performance and reduced computational requirements when employed as logical operators in a sorting-based preference ranking aggregation algorithm (Liu et al., 2024) that relies on logical coherence.

In sum, our contributions are as follows: **1)** We argue that logical consistency, in addition to human alignment, should be taken into account as an essential factor when aligning LLMs' preferences. We present a general mathematical formulation for quantifying/measuring three key properties of logical consistency: transitivity, commutativity and negation invariance. **2)** We then conduct extensive experiments to evaluate logical consistency across a range of state-of-the-art LLMs and analyze the correlations between consistency and models' reliability. **3)** We propose a data refinement and augmentation method for instruction-tuning that enhances logical consistency while maintaining alignment with human preferences and **4)** we show that LLMs with improved logical consistency yield improved performance in logic-dependent algorithms where LLMs are employed as logical operators. We will release our source code and refined dataset at ANONYMOUS to facilitate future work.

## 2  MEASURING LOGICAL CONSISTENCY

We evaluate the logical consistency of LLMs by assessing their ability to predict logically consistent relations among a *set* of *items*. These items could represent diverse *entities* or *events* with a uniform *relation* between them; such a relation might be (i) comparing the preference among response candidates to a query or (ii) determining the causal order of shuffled events, among other possibilities. This evaluation is grounded in relational algebra and order theory (Abiteboul et al., 1995; Imieliński & Lipski Jr, 1984), with the goal of assessing whether the model maintains coherent and self-consistent judgments across its predictions.

To formalize this concept, we define the *logical consistency evaluation process* by treating an LLM as an operator function that compares pairs of items and outputs a decision on the relation between the items. Let $X = \{x_1, x_2, \ldots, x_N\}$ represent a set of items, and we define a relation (e.g., comparison) function $F : X \times X \to R$, which compares two items, such as $(x_i, x_j)$, and assigns a relational decision $F(x_i, x_j) = r$, where $r \in R$ denotes the directional relation between $x_i$ and $x_j$. For simplicity, we consider $R$ to be a binary relation set, $R = \{r_{ij}, r_{ji}\}$, where $r_{ij}$ represents a preferential relation $x_i \succ x_j$ (i.e., item $x_i$ is preferred over item $x_j$), and $r_{ji}$ indicates the reverse preference $x_j \succ x_i$.

In evaluating logical consistency, we focus on whether the function $F$ adheres to the following key properties over the item set $X$: *1) transitivity*, *2) commutativity*, and *3) negation invariance*, as demonstrated in Figure 1. These properties are foundational for logically consistent operations and are critical for determining the reliability and reasoning capability of an LLM (Hamon et al., 2020).

Transitivity ensures that the LLM's predictions and judgements are internally coherent and do not suffer from logical contradictions within a given context. Commutativity tests whether the model's decisions are invariant to the order in which items are compared. The model should produce consistent judgments regardless of whether the pair $(x_i, x_j)$ or $(x_j, x_i)$ is presented first. The negation invariance checks whether the model maintains consistency when dealing with relational negations. Inconsistencies here would indicate a failure to correctly understand and apply the complementary nature of relations. By systematically applying these tests, we are able to assess the extent to which the model's outputs conform to logically consistent behavior, providing a quantitative proxy measure of its decision reliability.

### 2.1  MEASURING TRANSITIVITY

Grounded in our problem setup and definitions above, respecting transitivity implies the following: if a model predicts $A \succ B$ and $B \succ C$, it must also predict $A \succ C$. The ability to make transitive predictions is critical to forming a robust, global understanding of relationships within a set of items. Without transitivity, the model's reasoning would be prone to contradictions, which could lead to unreliable outcomes in tasks requiring consistent decision-making or ranking (Liu et al., 2024; Li et al., 2019; Qin et al., 2024).

We define that the function $F$ is fully transitive if it does not predict any intransitive relation within the set $X$. This means that if $F(x_i, x_j) = r_{ij}$ and $F(x_j, x_k) = r_{jk}$, then $F(x_i, x_k) = r_{ik}$ must hold for all $i, j, k \in X$. This can be visualized in Figure 2, where we represent the pairwise relations by $F$ as a relation graph. If $F$ is transitive over $X$, the corresponding relation graph should be a Directed Acyclic Graph (DAG). A DAG implies that *no cycles* exist in the relation graph, denoting that there are no contradictions in the model's relational judgments. Consequently, to determine if $F$ is fully transitive over the item set $X$, we only have to verify whether the predicted relation graph contains any cycle or not. We show how to construct the relation graph from the judgements of the LLM operator function $F$, and the algorithm to check whether a graph contains cycles in Appendix §A.

We introduce a metric, $s_{tran}(K)$, designed to quantify transitivity over an arbitrary number of items. LLMs often struggle to maintain perfect/full transitivity, especially as the number of items increases. The proposed metric $s_{tran}(K)$ captures the *degree of transitivity* across subsets of $K$ sampled items, where $3 \le K \le |X|$. The metric is defined as:

$$s_{tran}(K) = \frac{1}{M} \sum_{i=1}^{M} \mathbb{1}_{\text{acyclic}}(S_i^K).$$

(1)

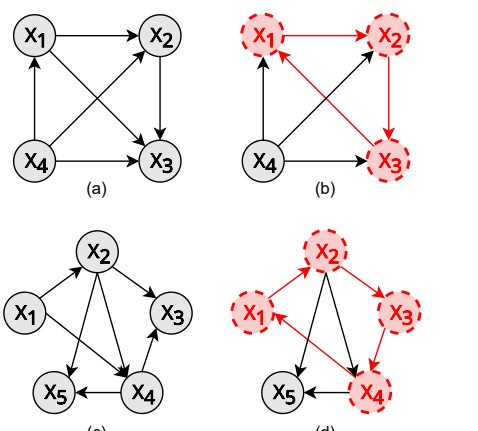 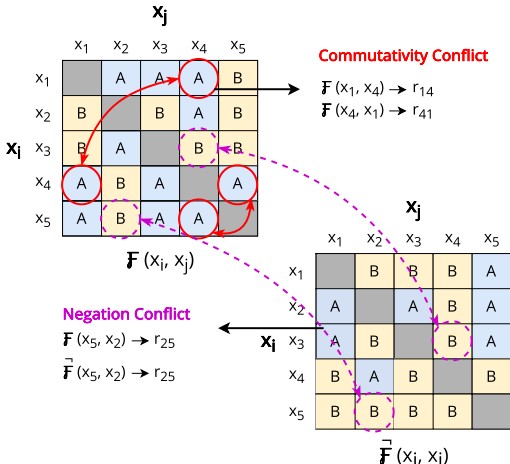

Figure 2: Example of relation graphs illustrating transitivity, where items are represented as nodes, and directed edges indicate pairwise preferential relations. Red dashed cycles in the graph highlight violations of transitivity. The cycle in (d), spanning 4 items, cannot be captured by $s_{tran}(3)$. The $s_{tran}$ metric can be applied to partial relation graphs, as shown in (c) and (d).

Figure 3: Examples illustrating violations of commutativity and negation invariance. Each entry of the two preference matrices represents predicted judgments of $x_i \succ x_j$ and $x_i \prec x_j$, labelled with A and B respectively. The top-left matrix is based on the original relation, while the bottom-right matrix reflects the negated relation. Linked red cycles highlight non-commutative pairs, and linked dashed purple cycles indicate negation inconsistencies.

Here, $S_i^K$ represents a randomly sampled sub-graph of size $K$, and the indicator function $\mathbb{1}_{acyclic}$ returns 1 if the sub-graph $S_i^K$ contains no cycles (i.e., is transitive), and 0 if otherwise. $M$ denotes the total number of the sampled sub-graphs. As the size of the item set increases, the number of possible sub-graph combinations grows exponentially. To manage this complexity, we cap the number of samples to 1,000 sub-graphs to estimate transitivity for larger sets. Therefore, the metric ranges from 0 to 1, where 1 represents perfect transitivity.

Maintaining transitivity becomes increasingly difficult as the size of the subset grows. For any set of $K$ items, there are $2^K$ possible combinations of pairwise relationships, but only $K!$ of these can form a transitive ranking. As a result, the degree of transitivity tends to decrease with larger sub-graph sizes. This means that for the same LLM, $s_{tran}(K)$ typically decreases as $K$ increases, reflecting that preserving consistent rankings over larger sets of items is increasingly challenging. Since $s_{tran}(K)$ measures transitivity for a fixed subset size, it allows for fair transitivity comparisons between item sets of different sizes.

## 2.2 MEASURING COMMUTATIVITY

Commutativity refers to the logical property that ensures the model's judgments remain consistent when the order of comparison between two items is reversed. Prior studies have shown that LLMs are susceptible to permutation bias, also referred to as positional bias (Wang et al., 2024b; Liusie et al., 2024a). This bias can result in inconsistent outputs when the order of items in a comparison is altered. To measure the degree of commutativity, we propose a metric $s_{comm}$, which evaluates whether the model's judgment changes when the order of the items is swapped in the prompt. Specifically, it is defined as follows:

$$s_{comm} = \frac{2}{|X| \cdot (|X| - 1)} \sum_{0 < i < j \leq |X|} \mathbb{1}(F(x_i, x_j) = F(x_j, x_i)). \tag{2}$$

Here, $F(x_i, x_j)$ represents the model's judgment when comparing items $x_i$ and $x_j$. The indicator function $\mathbb{1}$ returns 1 if the model's judgment remains consistent when the order of the items is reversed, i.e., $F(x_i, x_j) = F(x_j, x_i)$, and 0 otherwise. We visualize this comparison in Figure 3. The normalization term ensures that $s_{comm}$ is averaged across all pairwise combinations of the items in set $X$. As a result, the metric ranges from 0 to 1, with 1 indicating perfect commutativity, meaning that the model is completely robust to the order of item comparisons.

## 2.3 MEASURING NEGATION INVARIANCE

Negation invariance tests whether the model maintains consistency when confronted with the negation or inversion of a relational statement. Inconsistencies here could indicate a failure to correctly understand and apply the complementary nature of relations. Previous work suggested that LLMs struggle to automatically infer appropriate inverse relationships when acquiring knowledge (Allen-Zhu & Li, 2023; Berglund et al., 2024). To quantify negation invariance, we propose the metric $s_{neg}$, which examines if the model can correctly reverse its judgement when prompted with a negated relationship between items. The metric is defined as below:

$$s_{neg} = \frac{1}{|X| \cdot (|X| - 1)} \sum_{0 < i,j \leq |X|, i \neq j} \mathbb{1}(\overset{\neg}{F}(x_i, x_j) = \neg F(x_i, x_j)). \tag{3}$$

In this formulation, $\neg F(x_i, x_j)$ represents the negation of the original relation(e.g., reversing a preference or relational direction). The comparison function $\overset{\neg}{F}(x_i, x_j)$ refers to the model's judgment when explicitly prompted with the negated relation. The indicator function returns 1 if the model's response to the negated relation matches the expected negated judgment (i.e., $\overset{\neg}{F}(x_i, x_j) = \neg F(x_i, x_j)$), and 0 otherwise. We also visualize this comparison in Figure 3. The normalization factor averages across all pairwise permutations in the set $X$, ensuring that $s_{neg}$ ranges from 0 to 1. The maximum score of 1 indicates perfect negation invariance, where the model consistently handles negated relations.

## 3 EVALUATING LOGICAL CONSISTENCY OF LARGE LANGUAGE MODELS

After defining the measures quantifying the three aspects of logical consistency in Section 2, we now proceed to evaluate LLMs' judgements from the consistency angle on three representative tasks, each reflecting different levels of subjectivity.

### 3.1 EVALUATION SETUP

**Tasks and Datasets.** We employ three representative tasks to evaluate LLMs' logical consistency. The first task, *abstractive summarization evaluation*, uses the SummEval dataset (Fabbri et al., 2021) and focuses on the model's ability to make preference judgments between summaries, particularly regarding the coherence aspect. The second task, *document reranking*, leverages the NovelEval dataset (Sun et al., 2023), where LLMs assess the relevance of retrieved documents in response to queries. The final task, *temporal event ordering*, uses the CaTeRS dataset (Mostafazadeh et al., 2016b) and tests the model's capability to reason about temporal and causal relationships between events, critical for maintaining consistency in narrative understanding. Further task and dataset details are available in Appendix §B.

**Metrics and Reliability Measurement.** In Appendix §F, we show metric computation details and the prompt templates used. We compute the logical consistency metrics at the instance level and report averages across the test sets for each task. In addition, we report the human agreement rate (abbreviated as H.) by calculating the pairwise judgement accuracy between judgements made by LLMs and the provided human annotations. It serves as a reference for how closely the model's judgements align with human judgements. For the measurement of LLMs' reliability, we perform Monte Carlo Sampling of Chain-of-Thought (Wei et al., 2022b) reasoning outputs, using a temperature of 0.7. Self-agreement is defined as the percentage of outputs that agree with the majority judgment across multiple samples. This measurement ranges from 0.5 to 1, with higher values indicating greater stability in the model's reasoning processes.

### 3.2 RESULTS AND ANALYSIS

**Performance of Different (Families of) Models.** As shown in Table 1, recent LLMs like Gemma2-9B and Phi-3-medium demonstrate stronger overall consistency compared to earlier models. In particular, models such as Deepseek-chat, Phi-3-medium, and Gemma-2-9B perform well across all three evaluated consistency dimensions. However, it is important to note that strong performance in one aspect of consistency does not guarantee similar performance in others. For example, while Mistral-7B excels in transitivity, its performance in other consistency aspects is weaker.

Table 1: Logical consistency evaluation results. We report human accuracy (H.), transitivity over 5 items ($s_{tran}(5)$), commutativity ($s_{comm}$) and negation invariance ($s_{neg}$), all measured in accuracy.

| Models | SummEval (Coh) | | | | NovelEval | | | | CaTeRS | | | |
|---|---|---|---|---|---|---|---|---|---|---|---|---|
| | H. | $s_{tran}(5)$ | $s_{comm}$ | $s_{neg}$ | H. | $s_{tran}(5)$ | $s_{comm}$ | $s_{neg}$ | H. | $s_{tran}(5)$ | $s_{comm}$ | $s_{neg}$ |
| *Direct Judgements* | | | | | | | | | | | | |
| Llama-2-7B | 57.5 | 88.3 | 57.5 | 66.2 | 57.5 | 68.1 | 57.5 | 78.1 | 61.9 | 88.4 | 56.0 | 49.7 |
| Llama-2-13B | 58.3 | 86.6 | 59.3 | 84.0 | 58.3 | 88.2 | 62.9 | 76.9 | 65.6 | 95.3 | 70.8 | 54.5 |
| Llama-3-8B | 67.8 | 91.0 | 76.1 | 48.9 | 60.6 | 73.0 | 73.3 | 79.1 | 73.1 | 88.2 | 79.9 | 63.3 |
| Mistral-7B | 63.6 | 95.1 | 59.9 | 51.2 | 60.1 | 90.5 | 68.0 | 82.1 | 70.2 | 95.9 | 73.9 | 76.9 |
| Zephyr-7B-beta | 61.3 | 87.8 | 52.8 | 74.1 | 60.5 | 91.5 | 63.8 | 86.5 | 70.6 | 93.5 | 77.8 | 80.8 |
| Phi-3-mini | 65.6 | 92.8 | 66.9 | 75.1 | 59.7 | 92.5 | 55.5 | 39.4 | 60.2 | **97.2** | 85.5 | 73.3 |
| Phi-3-medium | 68.8 | **96.2** | 71.0 | 78.1 | 62.7 | 93.7 | 76.3 | 77.6 | 67.2 | 93.4 | 87.9 | 84.1 |
| Gemma-2-9B | **73.8** | 94.8 | **78.1** | 78.2 | 63.6 | 89.9 | 85.3 | 88.2 | 72.4 | 96.0 | 86.5 | 66.9 |
| GPT-3.5-0125 | 66.3 | 82.5 | 67.5 | 65.8 | 61.2 | 89.8 | 71.6 | 83.4 | 69.3 | 86.2 | 66.7 | 81.7 |
| Deepseek-chat | 72.7 | 93.1 | 72.8 | **84.9** | **65.7** | **95.1** | 86.7 | **89.2** | 73.5 | 93.8 | **91.9** | **91.1** |
| *CoT Prompting* | | | | | | | | | | | | |
| Llama-2-7B | 55.0 | 42.6 | 45.7 | 68.9 | 56.5 | 50.8 | 66.7 | 60.1 | 59.5 | 40.5 | 56.7 | 56.1 |
| Llama-2-13B | 66.6 | 55.8 | 63.2 | 40.3 | 59.2 | 60.8 | 71.0 | 56.4 | 60.6 | 61.1 | 63.5 | 74.3 |
| Llama-3-8B | 67.2 | 65.0 | 64.8 | 61.2 | 60.2 | 69.3 | 76.8 | 72.4 | 67.9 | 62.2 | 46.1 | 78.6 |
| Mistral-7B | 61.6 | 64.7 | 65.3 | 58.0 | 58.5 | 64.9 | 73.2 | 77.7 | 60.3 | 66.1 | 69.0 | 79.1 |
| Zephyr-7B-beta | 58.4 | 46.4 | 51.7 | 74.0 | 60.0 | 64.5 | 70.6 | 71.2 | 65.7 | 69.3 | 78.4 | 77.3 |
| Phi-3-mini | 65.1 | 67.9 | 61.9 | 43.0 | 56.5 | 61.2 | 47.8 | 80.5 | 58.4 | 81.2 | 82.6 | 78.7 |
| Phi-3-medium | 69.1 | **82.3** | 72.5 | 58.9 | **63.1** | 89.0 | 87.3 | 79.3 | 54.5 | 81.4 | 87.4 | 85.9 |
| Gemma-2-9B | 73.5 | 74.0 | 71.9 | 81.3 | 62.1 | 89.9 | 85.9 | 82.3 | 63.5 | **83.8** | 87.1 | 69.9 |
| GPT-3.5-0125 | 63.9 | 63.3 | 64.8 | 68.4 | 59.3 | 79.3 | 73.4 | 67.5 | 62.8 | 68.6 | 70.5 | 74.9 |
| Deepseek-chat | **74.9** | 79.4 | **76.1** | 82.8 | 61.5 | **94.2** | 90.4 | 83.0 | 69.1 | 81.6 | **88.2** | **88.4** |

The Phi-3 family consistently shows high logical consistency, which we hypothesize is due to its reliance on synthesized data which is cleaner and less self-contradictory. This type of data likely reduces internal conflicts, resulting in more consistent logical behavior. However, no strong correlation is observed between logical consistency and human agreement accuracy. For instance, while Gemma-2-9B and Llama-3-8B achieve similar levels of human agreement, Gemma-2-9B shows higher consistency in its outputs. This is further confirmed by the low correlations (below $0.05$) between transitivity consistency and human preference accuracy across all datasets.

Another observation is that LLMs tend to perform more consistently on the CaTeRS dataset. We attribute this to the dataset's focus on temporal and causal relations. The more objective and logical preferential relations may make it easier for models to maintain consistency.

**Impact of CoT Prompting.** We also investigated the effect of CoT prompting on logical consistency. Surprisingly, CoT reasoning did not generally improve consistency, and in some cases, it led to a decrease in transitivity performance. We hypothesize that this decline may be due to the additional reasoning tokens generated during CoT prompting, which could introduce variations in the judgement standards used for different comparisons. When a model's understanding of the preferential relations is not uniform, it may produce inconsistent (non-transitive) outcomes. This suggests that CoT prompting, while beneficial for complex reasoning, might introduce unintended inconsistencies in certain logical judgement tasks.

### 3.3 CONSISTENCY AND RELIABILITY

**Transitivity as a Proxy for Self-Agreement.** We next investigated the relationship between transitivity and self-agreement across different LLMs. As shown in Figure 4, there are strong correlations between transitivity and self-agreement across all three datasets, regardless of the task's level of subjectiveness. Self-agreement indicates the model's internal consistency and robustness, as a reliable model should not fluctuate significantly in its responses. Higher self-agreement suggests that the model exhibits a stable understanding of the underlying relations in the input. Given that transitivity captures this self-consistency, we argue that transitivity serves as a useful proxy for evaluating the local robustness of LLMs.

**Commutativity Correlates Strongly with Human Preferences.** For each task, we paraphrased the comparison prompt template 10 times using GPT-4-turbo to explore the sensitivity of LLMs to

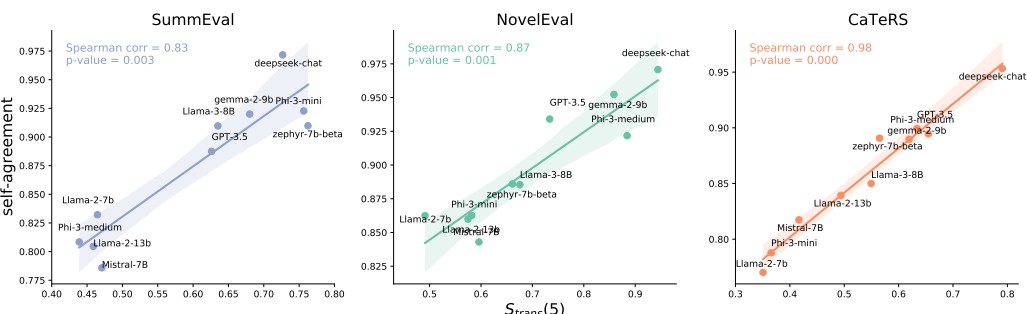

Figure 4: *Transitivity shows strong correlations with self-agreement* across all three datasets. Self-agreement is measured as the percentage of majority choices from 10 CoT inferences, generated with a temperature of 0.7.

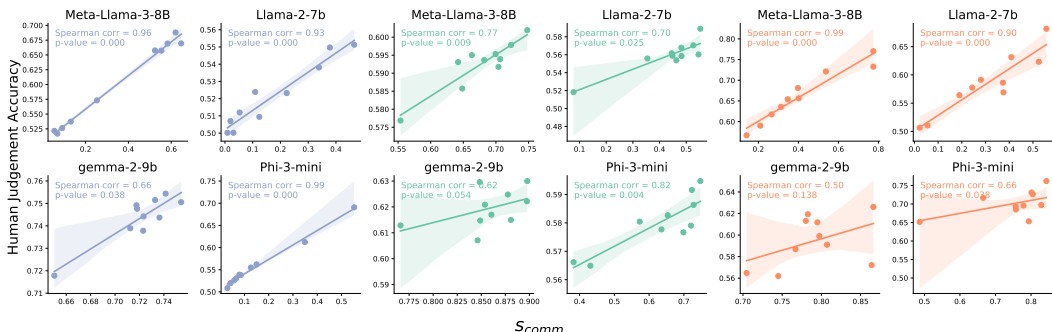

Figure 5: *Commutativity shows a generally strong correlation with human preference agreement rates* across various LLMs and all three datasets.

input variations. Despite these paraphrases being semantically equivalent, they produced different performance outcomes, as shown in Figure 5.

We found strong correlations between commutativity and human agreement rates across nearly all datasets and models. This finding is not surprising, as a lack of commutativity often indicates a strong positional bias. These results align with previous research (Zhou et al., 2024), which demonstrated that positional bias can significantly impact alignment with human judgments.

We also show that the strength of this correlation varies between models. For instance, Llama-3-8B exhibits a higher correlation with human preferences compared to Gemma-2-9B. We hypothesize that this difference is due to Gemma-2-9B's training, which was designed to be more robust to input prompts, while other models, like Llama-3-8B, are more fine-tuned to specific prompting styles.

# 4 A DATA REFINEMENT AND AUGMENTATION APPROACH TO IMPROVE LOGICAL CONSISTENCY IN LLMS

Our analysis so far has demonstrated that many LLMs exhibit varying degrees of logical (in)consistency when making judgements. To address this challenge, we propose a data refinement and augmentation framework that aims to mitigate this phenomenon. The idea is to filter out self-conflicting information from noisy preference data and generate additional conflict-free pairwise comparisons. This approach aims to enhance the logical consistency of LLMs while maintaining alignment with human preferences, enabling them to function more reliably as logical operators.

**Motivation.** Previous work has proven the benefits of using pairwise comparisons derived from preference rankings to train LLMs for a better understanding of preference orderings (Song et al., 2024). Additionally, Asai & Hajishirzi (2020) showed that incorporating logically derived counterparts of

Figure 6: *Illustration of data refinement and augmentation* based on preference rank estimation using win-loss rates. The original annotations (left) are sparse and exhibit inconsistencies (non-transitivity), while the augmented annotations are consistent and include *more* comparisons. For partially ordered rankings, items are not compared when their preference relationship is unknown.

Table 2: Data statistics at each augmentation stages. # Data refers to data size and AvgComp/Inst for average comparisons per instance. Aug. data represents Augmented data, and Neg. denotes further augmentation with negated relation comparisons.

| Data | $s_{tran}(5)$ | $s_{comm}$ | # Data | AvgComp/Inst |
|---|---|---|---|---|
| Raw data | 98.4 | - | 14.8K | 6.29 |
| Perturbed data | 87.6 | - | 14.8K | 6.29 |
| Aug. data | 1 | 1 | 30.9K | 13.2 |
| Aug. data+Neg. | 1 | 1 | 61.8K | 26.4 |

Table 3: Instruction-tuning (IT) of Llama-3 Instruct (8B) using different dataset variants (see Section 4.2). Only training on negated relations can enhance negation invariance.

| Models | Summarize from Feedback | | | |
|---|---|---|---|---|
| | H. | $s_{tran}(5)$ | $s_{comm}$ | $s_{neg}$ |
| Zero-shot inference | 64.3 | 81.1 | 70.2 | 63.8 |
| IT Perturbed data | 70.3 | 91.9 | 88.4 | 61.0 |
| IT Augmented data | 70.1 | 95.4 | 91.2 | 60.8 |
| IT Augmented data + Neg. | 69.7 | 95.2 | 90.5 | 87.9 |

pairwise annotations improves consistency in knowledge-intensive QA tasks. However, real-world preference data are often noisy and self-contradictory (Chowdhery et al., 2022; Ouyang et al., 2022a), especially in more subjective tasks (Bai et al., 2022). The inter-rater agreement rate typically falls between $60\%$ to $80\%$ for preference or evaluation datasets. Prior work (Wang et al., 2024a) has shown that these self-conflicting annotations reduce the efficiency of learning preferences, and we hypothesize that such annotations also contribute to the logical inconsistencies in trained models.

Inspired by the RLHF process, where a reward model is used to establish a complete order of responses, we propose first estimating a coherent ranking from the noisy data. This refined ranking can then be augmented with additional conflict-free pairwise comparisons to align LLMs more effectively. This method is expected to be more efficient than aligning models to noisy and inconsistent annotations directly.

## 4.1 ESTIMATING RANKINGS FROM NOISY PAIRWISE ANNOTATIONS USING WIN-LOSS RATE

Estimating global rankings from noisy pairwise annotations is essentially a rank aggregation problem. We use the win-loss rate method due to its simplicity and two key advantages: **1)** it performs well with partial and sparse comparisons, which is common in real-world preference datasets, and **2)** it remains unaffected by the order in which comparisons are presented. While other rank aggregation methods like ELO or the Bradley-Terry model (Bradley & Terry, 1952) could be explored for improved results, our choice serves as a proof-of-concept rather than an optimization for best performance.

The win-loss rate for each item is calculated as the number of comparisons it wins minus the number of comparisons it loses and then divided by the number of comparisons it participates in. Following that, as shown in Figure 6, we rank the item by the value of its win-loss rate. This way, we can aggregate a full or partial ranking. We then split the ranking into a self-consistent pairwise comparison set. We can further augment the dataset by adding the comparisons using the negated relation.

Table 4: LLMs with better transitivity perform better with PairS. Improved commutativity indicates less need for algorithm calibration. For both PairS methods, we report the average Spearman correlations over 100 runs.

| Models | SummEval (Coh) | | | PairS | PairS calibrated |
|---|---|---|---|---|---|
| | H. | $s_{tran}(5)$ | $s_{comm}$ | | |
| Mistral-7B | 63.6 | 95.1 | 59.9 | 27.7 | 31.2 +3.5 |
| Phi-3-mini | 65.6 | 92.8 | 46.9 | 33.9 | 38.0 +4.1 |
| GPT-3.5-turbo | 66.3 | 82.5 | 67.5 | 33.5 | 36.3 +2.8 |
| Llama-3-8B | 67.8 | 91.0 | 76.1 | 37.7 | 38.9 +1.2 |
| Phi-3-medium | 68.8 | 96.2 | 71.0 | 38.9 | 41.3 +.2.4 |

## 4.2 EXPERIMENTS

**Experimental Setup.** We use the Summarize From Feedback dataset (Mostafazadeh et al., 2016b), where human annotators made pairwise comparisons between two summaries based on qualitative factors. More details are available in Appendix §B. Given that the dataset annotations are sparse and relatively clean, we simulate noise by randomly flipping $10\%$ of the training labels. Our data refinement and augmentation techniques are then applied to improve the consistency and quantity of pairwise comparisons, as shown in Table 2.

To assess the effectiveness of our approach, we instruction-tuned three Meta-Llama-3-8B-Instruct models on: **1)** the flipped/perturbed data, **2)** the refined and augmented dataset, and **3)** the 'further augmented' dataset with additional negated relation comparisons. Training parameters are detailed in Appendix § E. We randomly sample 200 instances from the test set and evaluate all models on this subset. We assess the impact of the data augmentation on logical consistency and performance.

**Results and Findings.** The results shown in Table 3 highlight three key findings: **1)** Zero-shot inference shows considerable logical inconsistency. However, training on perturbed data can already substantially improve both human preference alignment and logical consistency. **2)** Training with the refined and augmented dataset significantly improves transitivity and commutativity, while maintaining strong human alignment. **3)** Only training further on negated relations improves the model performance in negation invariance. However, adding negated relations to the broader dataset may introduce distractions and cause a forgetting effect (Luo et al., 2023), resulting in a slight reduction in performance on other logical properties. Overall, the findings support the effectiveness of our data refinement and augmentation framework in improving the logical consistency of LLMs.

## 5 IMPACT OF LOGICAL CONSISTENCY ON DOWNSTREAM APPLICATIONS

LLMs are increasingly used as logical operators in high-level algorithms due to their ability to process and understand text semantics. For instance, Qin et al. (2024) use LLMs to enhance document reranking in information retrieval systems. Similarly, Guo et al. (2024) and Yang et al. (2024) utilize LLMs as optimizers in prompt-tuning algorithms, while Liu et al. (2024) and Liusie et al. (2024b) employ LLMs as pairwise comparators to aggregate preference rankings. When LLMs are used as logical operators, maintaining a high level of logical consistency is critical to ensure predictable and efficient decision-making. In this section, we examine how logical consistency influences the performance of LLM-based algorithms in such 'logically grounded' tasks.

### 5.1 APPLICATION EXAMPLE: PREFERENCE RANKING ALGORITHM

We illustrate the impact of logical consistency through the Pairwise-Preference Search (PairS) algorithm proposed by Liu et al. (2024). PairS is a sorting-based rank aggregation method that uses LLMs as pairwise evaluators (i.e., it is a particular *'LLM-as-a-judge'* algorithm), comparing items based on specific attributes using a merge-sort approach. Its performance is measured by comparing the LLM-generated rankings with human judgments via Spearman's correlation. Sorting algorithms depend heavily on logical properties such as transitivity and commutativity. PairS assumes that LLMs used as evaluators have near-perfect transitivity and commutativity for optimal ranking results.

To evaluate this, we conduct controlled experiments on the coherence aspect of SummEval, using various LLMs with similar accuracy to human judgments. The results are shown in Table 4. As LLMs

often exhibit positional bias, we follow Liu et al. (2024) by reporting both raw and calibrated results. The calibration, performed as described in Zheng et al. (2023), averages the evaluation probabilities across both possible pairwise orders. While this technique increases inference cost, it reduces bias by balancing positional preferences.

Our findings from Table 4 reveal two key insights: **1)** Although Phi-3-mini has slightly lower accuracy with human judgments than GPT-3.5-turbo, its stronger transitivity leads to better ranking performance with or without calibration. **2)** There is a clear correlation between an LLM's commutativity and the performance gains from calibration in the PairS algorithm. LLMs that are more commutative rely less on calibration to achieve optimal performance, and therefore, require less computations.

# 6 RELATED WORK

Consistency of LLMs has been explored in several contexts before, where previous research has predominantly focused on two domains: *consistency of factual knowledge* and *entailment consistency* across limited statements.

**Consistency of Factual Knowledge.** Previous work has demonstrated that in knowledge-intensive question-answering tasks, concepts like symmetry and transitivity of knowledge snippets are important (Asai & Hajishirzi, 2020). To this end, a benchmark for studying language model consistency with paraphrased relations (Elazar et al., 2021) has been created. Additionally, different studies have examined whether LLMs have a coherent understanding of knowledge graphs related to real-world entities (Jung et al., 2022; Gu et al., 2023; Kassner et al., 2023; Gu et al., 2023). The 'reverse curse' phenomenon highlights that unlearned factual knowledge cannot be deduced by reversing learned knowledge (Allen-Zhu & Li, 2023; Berglund et al., 2024).

**Entailment and Inference Consistency.** In Natural Language Inference (NLI), the consistency of transitivity and symmetry in statement pairs has been explored using predefined inference rules (Li et al., 2019; Jang & Lukasiewicz, 2023). Jang et al. (2022) proposed a test set to exam the consistency of LLMs for NLI tasks. Prediction consistency has been improved through the use of adversarial first-order logic examples (Minervini & Riedel, 2018).

Despite the wealth of knowledge regarding the logical consistency of LLMs within specific domains, most studies have concentrated on first-order relations—logical connections or implications directly linking two or three individual statements. Consequently, there is a notable gap in research addressing the consistency and reliability of LLMs when applied to more complex decision-making scenarios or the evaluation of multiple items simultaneously. This limitation suggests a pressing need for further investigation into how LLMs can maintain coherence and consistency in broader, more comprehensive contexts, which is essential for their deployment in practical applications.

# 7 CONCLUSION

In this work, we investigated the critical role of logical consistency in enhancing the reliability and trustworthiness of LLMs, especially in their roles as decision-makers and reasoners. We introduced a general framework for quantifying three key properties of logical consistency: transitivity, commutativity, and negation invariance. Through comprehensive evaluation, we demonstrated that these consistency measures serve as strong proxies for assessing the overall reliability of LLMs.

To improve the logical consistency of LLMs, we then proposed a data refinement and augmentation framework, which effectively reduces logical inconsistencies at the source: directly in the data. Our experimental results showed that models trained on this refined and augmented data achieve improved logical consistency without compromising alignment with human preferences. Furthermore, we demonstrated that integrating LLMs with higher logical consistency into a logic-driven algorithm improves both the algorithm's global performance and computational efficiency. This highlights the practical benefits of ensuring logical coherence in LLMs when applied to downstream tasks.

In summary, our work emphasizes the importance of logical consistency as a complementary factor to human alignment in the development of more reliable LLMs. We hope this research will encourage the broader community to recognize the importance of logical consistency as a fundamental aspect of any work aiming towards improved trustworthiness and reliability.

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

## A    TRANSITIVITY CALCULATION

In this section, we show two algorithms: (i) construct a preference relation graph by LLM's judgements, and (ii) detect cycle in a relation graph.

---

**Algorithm 1** Construct Preference Relation Graph

---

1: **Objective:** Construct a relation graph reflecting the understanding of LLM upon a set of items.
2: **Inputs:** $\mathbf{X} = \{x_1, x_2, \ldots, x_N\}$: A set of items; $\text{LLM}(\cdot, \cdot | I)$: LLM, for a given instruction prompt $I$, as a preference operator.
3: **Output:** $G$: A relation graph represented as an adjacency list for each vertex.
4: **Initialize:** Initialize the relation graph $G$ such that for each $x$ in $X$, $G[x] \leftarrow \varnothing$, where $G[x]$ represents the pointed vertexes from vertex $x$.
5: **for** $i \leftarrow 1$ **to** $n$ **do**
6:     **for** $j \leftarrow i + 1$ **to** $n$ **do**
7:         $P(x_i \succ x_j) \leftarrow \text{LLM}(x_i, x_j | I)$
8:         **if** $P(x_i \succ x_j) \geq \frac{1}{2}$ **then**
9:             $G[x_i] \leftarrow G[x_i] \cup \{x_j\}$
10:         **else**
11:             $G[x_j] \leftarrow G[x_j] \cup \{x_i\}$
12:         **end if**
13:     **end for**
14: **end for**
15: **Return** $G$

---

**Algorithm 2** Check for Cycles in a Directed Graph

---

1: **Objective:** Detect if a directed graph $G$ contains a cycle.
2: **Inputs:** $G$: A directed graph represented as an adjacency list.
3: **Output:** True if a cycle exists, False otherwise.
4: **Initialize:** $visited \leftarrow \varnothing, recStack \leftarrow \varnothing$
5: **for** each $v$ in $G$ **do**
6:     **if** $v \notin visited$ **and** CycleUtil($v$, visited, recStack) **then**
7:         **Return** True
8:     **end if**
9: **end for**
10: **Return** False
1: **function** CYCLEUTIL($v$, visited, recStack)
2:     $visited \leftarrow visited \cup \{v\}, recStack \leftarrow recStack \cup \{v\}$
3:     **for** each $u$ in $G[v]$ **do**
4:         **if** $u \notin visited$ **and** CycleUtil($u$, visited, recStack) **then**
5:             **Return** True
6:         **else if** $u \in recStack$ **then**
7:             **Return** True
8:         **end if**
9:     **end for**
10:     $recStack \leftarrow recStack \setminus \{v\}$
11:     **Return** False
12: **end function**

---

## B    DATASET

**SummEval** (Fabbri et al., 2021) is a summarization meta-evaluation dataset comprising 100 source texts, each paired with 16 summary candidates generated by various language models (LMs). The dataset is annotated based on four criteria: coherence (COH), fluency (FLU), consistency (CON), and relevancy (RE).

**NovelEval** (Sun et al., 2023) is a document reranking test set consisting of 21 novel questions. This set was constructed by compiling questions and passages from four domains published after the release of GPT-4. For each question, 20 candidate passages were retrieved through Google search, and their relevance was manually labeled.

**CaTerS** (Mostafazadeh et al., 2016b) is a dataset focused on temporal event ordering. It contains 1,600 annotated sentences derived from 320 five-sentence short stories sampled from the ROCStories corpus (Mostafazadeh et al., 2016a). These stories are organized based on causal and temporal relations. We filtered the dataset to include only instances with seven or more events, resulting in 70 instances.

**Summarize from Feedback.** (Mostafazadeh et al., 2016b) consists of 64,832 summary comparisons drawn from the TL;DR (Cachola et al., 2020) and CNN/DM datasets (Nallapati et al., 2016). This dataset is divided into two sections: pairwise comparisons and axis annotations. We concentrate on the pairwise comparison section, which is labeled based on the overall quality of two summaries. For training, we utilize the entire training set, while for testing, we randomly sample 200 instances from the validation set.

**MS MARCO.** (Nguyen et al., 2016) is a widely used large-scale benchmark for training and evaluating retrieval-based question answering systems. Following the methodology of Sun et al. (2023), we leverage the dataset for a document reranking task. Specifically, they randomly sample a subset of queries from the MS MARCO training set, retrieving 10 candidate passages for each query using BM25. Ranking preferences for these passages were distilled using GPT-3.5-turbo. In this work, we randomly sample 1K examples from their query subset and corresponding ranking labels. We then split them into a training set of 800 examples and a test set of 200 examples.

## C  LARGE LANGUAGE MODELS IN EVALUATION

**Open-Sourced LLMs.** The Llama-2-7b-chat-hf and Llama-2-13b-chat-hf models (Touvron et al., 2023) were developed by Meta and are part of the Llama-2 family. Llama-3-8B-Instruct (Dubey et al., 2024) is another model from Meta's Llama series. The Mistral-7B-Instruct-v0.1 (Jiang et al., 2023) model is an instruction-tuned LLM developed by Mistral AI. The zephyr-7b-beta model (Tunstall et al., 2023) is developed by Zephyr AI. The Phi-3-mini-4k-instruct and Phi-3-medium-4k-instruct models (Abdin et al., 2024) are part of the Phi model series developed by Microsoft. The gemma-2-9b-it (Rivière et al., 2024) model is created by the Gemma team.

**API-Based LLMs.** The GPT-3.5-turbo-0125 model is part of OpenAI's GPT-3.5 series, which is renowned for its strong performance in natural language understanding and generation. Deepseek-chat (DeepSeek-AI, 2024) is another API-based LLM developed by DeepSeek.

## D  ADDITIONAL RELATED WORK

**Improving Consistency of LLMs.** Asai & Hajishirzi (2020) proposed an augmentation method and a training regularization term to improve the logical consistency of language models for QA tasks. Minervini & Riedel (2018) proposed an adversarial training procedure to improve the robustness of NLI models to adversarial examples. Wang & Henao (2021) focus on improving the paraphrasing consistency during training for the task of low-resource Named Entity Recognition (NER).

Kumar & Joshi (2022) performed a symmetric consistency analysis on NLI and semantic textual similarity (STS) tasks in a more conservative manner, arguing that a model should generate not only the same predictions but also the same confidence scores if it is truly input-order invariant. They also observed that pretrained language models violated symmetric consistency and the authors then introduced a consistency regularisation term to compensate for the issue.

**Theory of Consistency.** The concept of transitivity in preferences has been extensively explored, revealing its significance in both psychological and logical frameworks. Tversky (1969) highlighted various psychological factors that can lead to intransitive preferences, illustrating how human decision-making often deviates from rational models. Building on this, Arrow (1950) introduced Arrow's Impossibility Theorem, which addresses the inherent challenges of constructing social choice functions that maintain transitivity.

Further contributions to the understanding of transitivity include a comprehensive review by Regenwetter et al. (2011), which examines both theoretical and empirical perspectives on the subject. Their work underscores the critical role transitivity plays in preference structures. In a more formal context, Tarski (1941) examined the role of transitivity within relational calculus, an area that is foundational to

many logical systems. This connection demonstrates how transitivity not only influences preferences but also underpins logical reasoning.

Moreover, Hansson (2001) provided insights into the implications of violating transitivity in preferences, showing that such violations can lead to logical inconsistencies within decision theory. This theme is further explored by Hyde (2011), who proved that apparent violations of transitivity can give rise to logical paradoxes. Collectively, these works emphasize the importance of transitivity in ensuring consistency, both in decision-making and logical deduction.

## E    INSTRUCTION-TUNING: TRAINING DETAILS

The model was fine-tuned using the following hyperparameters. A learning rate of $5 \times 10^{-5}$ was employed over the course of 2 training epochs. A weight decay of $1 \times 10^{-2}$ was applied to prevent overfitting. For the LoRA (Low-Rank Adaptation) settings, the rank $r$ was set to 16 and the scaling factor $\alpha$ was set to 64. The batch size during training was 4, with a gradient accumulation step of 2 to effectively handle smaller batches. All training was performed on an A100 machine.

## F    METRIC COMPUTATION DETAILS AND PROMPT TEMPLATES

**Metrics Computation details.** For all three datasets, they have the format of each input context is associated with multiple items, and the task is to determine specific relations among them.

For **SummEval**, the input context is a source text, and the task involves determining coherence preferences among multiple summary candidates. In **NovelEval**, the input context is a query, with the task focused on assessing relevance preferences among multiple retrieved documents. For **CaTeRS**, the input context consists of a list of unordered events, with the goal being to infer the temporal or causal order among these events.

To compute the consistency metrics, a full pairwise comparison is conducted between every pair of items, including all permutations of the item set $X$. Both standard expression ($F(x_i, x_j)$) and its reverse ($\overline{F}(x_i, x_j)$) are used. This will result two preference matrices as shown in Fig. 3, and can be used to compute the $S_{tran}$, $S_{comm}$ and $S_{neg}$ following the Equ. 1, Equ. 2 and Equ. 3.

**Prompt Templates.** we demonstrate the prompt templates we used to perform pairwise comparisons for all three datasets in Figure 7. Figure 8 illustrates the prompt templates utilized for conducting the instruction tuning experiments on the Summary with Feedback dataset, as detailed in Section 4.

## G    THE CHOICE OF K

In this section, we discuss the considerations regarding the choice of $K$ for the transitivity metric $S_{tran}(K)$.

**Robustness of transitivity to $K$.** $S_{tran}(K)$ is robust to variations in $K$. It is uncommon for a model to perform well on $S_{tran}(3)$ but poorly on $S_{tran}(5)$, as shown in the Figure 9. Additionally, it shows that larger values of $K$ expand the value range of $S_{tran}(K)$, making comparisons between models more distinct.

**Dependence on Prior Knowledge.** Choosing $K$ can depend on prior knowledge about the general consistency performance of current LLMs. For example, we observe that the consistency performance improves progressively from earlier LLMs, such as LLama-2 and Zephyr-7B-beta, to more advanced models like Gemma-2 and Llama-3. This trend suggests that considering the evaluated LLMs' expected consistency levels can inform an appropriate choice of $K$.

**Task-Specific Considerations.** As discussed in Section 3.2, consistency performance varies depending on the task. For objective tasks, consistency tends to be higher, which means task-specific nuances should also guide the selection of $K$.

In summary, there is no definitive "gold standard" for choosing $K$. Similar to selecting the N-gram size in BLEU-$N$, the choice depends on the models and tasks under evaluation.

**Prompt for F $(x_1, x_2)$**

```
"""
Source Text: {{ input }}

Summary candidate A: {{ output_1 }}
Summary candidate B: {{ output_2 }}

Question: Which summary
candidate has better coherence?
Please only answer with A or B.

Answer:
"""
```

**Prompt for F $(x_2, x_1)$**

```
"""
Source Text: {{ input }}

Summary candidate A: {{ output_2 }}
Summary candidate B: {{ output_1 }}

Question: Which summary
candidate has better coherence?
Please only answer with A or B.

Answer:
"""
```

**Prompt for $\overline{\text{F}}$ $(x_1, x_2)$**

```
"""
Source Text: {{ input }}

Summary candidate A: {{ output_1 }}
Summary candidate B: {{ output_2 }}

Question: Which summary
candidate has worse coherence?
Please only answer with A or B.

Answer:
"""
```

(a) SummEval

**Prompt for F $(x_1, x_2)$**

```
"""
Query: {{ input }}

Document candidate A: {{ output_1 }}
Document candidate B: {{ output_2 }}

Question: Which document candidate
is more relevant to the query?
Please only answer with A or B.

Answer: """
```

**Prompt for F $(x_2, x_1)$**

```
"""
Query: {{ input }}

Document candidate A: {{ output_2 }}
Document candidate B: {{ output_1 }}

Question: Which document candidate
is more relevant to the query?
Please only answer with A or B.

Answer: """
```

**Prompt for $\overline{\text{F}}$ $(x_1, x_2)$**

```
"""
Query: {{ input }}

Document candidate A: {{ output_1 }}
Document candidate B: {{ output_2 }}

Question: Which document candidate
is less relevant to the query?
Please only answer with A or B.

Answer: """
```

(b) NovelEval

**Prompt for F $(x_1, x_2)$**

```
"""
Event context:
{{ input }}

Question: Does {{ output_1 }} happen
temporally or causally later than
{{ output_2 }}?
Please only answer with 'Yes' or 'No'.

Answer:"""
```

**Prompt for F $(x_2, x_1)$**

```
"""
Event context:
{{ input }}

Question: Does {{ output_2 }} happen
temporally or causally later than
{{ output_1 }}?
Please only answer with 'Yes' or 'No'.

Answer:"""
```

**Prompt for $\overline{\text{F}}$ $(x_1, x_2)$**

```
"""
Event context:
{{ input }}

Question: Does {{ output_1 }} happen
temporally or causally earlier than
{{ output_2 }}?
Please only answer with 'Yes' or 'No'.

Answer:"""
```

(c) CaTeRS

Figure 7: Prompt templates for pairwise comparisons. From left to right: Normal comparison, comparison with reversed item order, and comparison with negated relation.

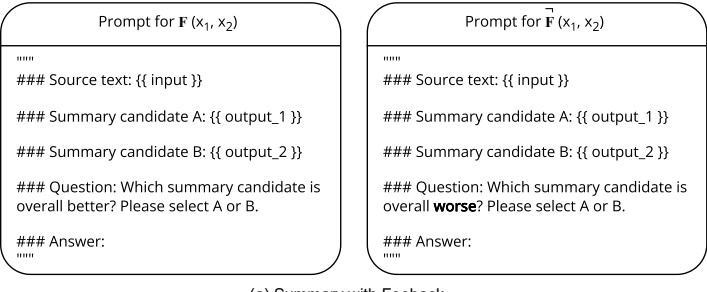

(a) Summary with Feeback

Figure 8: Pairwise comparison prompt templates for the instruction tuning on *Summary with Feedback*, as discussed in Section 4.

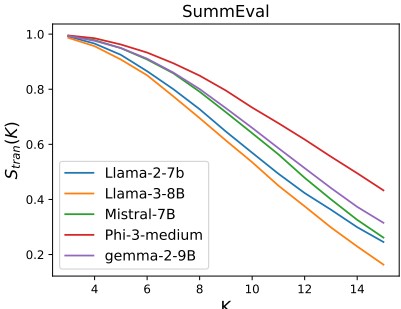

Figure 9: Transitivity metric $S_{tran}(K)$ as a function of sub-graph size, $K$, on the SummEval dataset. Transitivity values decrease as $K$ increases.

## H   THEORETICAL EVIDENCE FOR THE CHOICE OF TRANSITIVITY SUB-GRAPH SAMPLING SIZE $M$

**Theoretical Basis.** The transitivity metric $S_{tran}(K)$ is computed as the average result of a binary indicator function (as defined in Equation 1), which follows a binomial distribution. Intuitively, $S_{tran}(K)$ can be interpreted as the probability ($p$) that a randomly sampled $K$-node sub-graph is transitive.

**Estimation Accuracy.** Using the Central Limit Theorem, the binomial distribution of $S_{tran}(K)$ approximates a normal distribution when the sample size ($n$) is sufficiently large. To estimate $p$ within a margin of error ($E$) at a confidence level $z$ (e.g. $z = 1.96$ for $95\%$ confidence), the required sample size is:

$$n = \frac{z^2 \cdot p \cdot (1 - p)}{E^2} \tag{4}$$

**Worst-Case Scenario Analysis.** Since the exact value of $p$ is unknown, we consider the worst-case scenario where $p(1 - p)$ is maximized at $p = 0.5$. Under this assumption, for a margin of error $E$ at $95\%$ confidence ($z = 1.96$). The sample size $n$ should be at least 394, which is far exceeded by our choice of $M = 1000$. Therefore, our estimation of the $S_{tran}(K)$ is statistically accurate.

