# OpenReview forum: "Measuring, Evaluating and Improving Logical Consistency in Large Language Models"
_ICLR.cc/2025/Conference — Submitted to ICLR 2025_

### Official Review · Reviewer_rqBr · 2024-11-01

**Soundness:** 3
**Presentation:** 2
**Contribution:** 2
**Rating:** 3
**Confidence:** 3

**Summary:**

This paper shows an empirical study of the consistency of the LLMs using 3 metrics, transitivity, commutativity, and negation. This is an empirical work that evaluates many (10 models) over 4 datasets.
In Section 4, the paper also suggests to use of data augmentation with a rank estimation (win-loss rates) for improving the consistency performance.

**Strengths:**

The paper defines 3 metrics and evaluates on large number of models over 4 datasets. Overall results are helpful to understand the current status of LLMs. For example, if we see $S_{comm}$ the scores are low in many cases which is consistent with findings from other papers. It is also good to see the results from CoT prompting that degrades the consistency performance.

**Weaknesses:**

This paper mostly explores the empirical behavior of LLMs given 3 performance metrics, and it mentions possible usage for fine-tuning. However, we don't see how to leverage the findings. The evaluation sets are small, and it is not sure if the findings will be applicable in general. The datasets also come with "candidates" to select. That means LLMs only need to select an answer from candidates. It is not clear if this connects to the concept of logical consistency.

**Questions:**

The measures depend on the sampling. How do the scores vary? What are the items ($x_i$ or $x_j$) that we put in the equations?

Are the experiments only conducted in multi-choice datasets that require LLMs to rank the candidates available?

Section 4 needs more explanations. The paper doesn't show the details about the instruction-tuning settings.

---

> ### Author Response · Authors · 2024-11-22
> **Response to the reviewer (1)**
>
> We sincerely thank the reviewer for their time and effort in reviewing our paper. We greatly appreciate the insightful feedback provided, as we believe it will significantly contribute to improving the quality of our work. Below, we address and clarify the concerns raised:
>
> ---
>
> > W1: This paper mostly explores the empirical behavior of LLMs given 3 performance metrics, and it mentions possible usage for fine-tuning. However, we don't see how to leverage the findings.
>
> We would like to address the potential misunderstanding of our paper’s scope and contributions. **Our work systematically examines logical consistency in LLMs from both interpretative and application-oriented perspectives**. We believe these contributions, detailed below, provide both theoretical and practical insights:
>
> 1. **Interpretation and Analysis** (Sections 2 and 3):
>     - We **introduce three novel metrics** (Commutativity, Transitivity, and Negation Invariance) to measure logical consistency in LLMs. These metrics are rigorously formalized, allowing us to evaluate LLM behavior across a wide range of models and settings.
>       - Unlike the existing benchmarks, *our metrics are not designed to assess accuracy in knowledge or reasoning but provide a deeper lens into the logical consistency within LLMs*. This information is vital for providing guidance to downstream applications where consistency, not just correctness, is crucial.
>       - Furthermore, we demonstrate how our metrics correlate with reliability measures, offering a new dimension to evaluate LLM behavior.
>
>    - Each metric addresses a distinct application need:
>      - **Commutativity** helps address positional bias when making judgements.
>      - **Transitivity** is essential in applications that operate on local comparisons but a reliable global ranking is desired to maintain, such as sorting and optimization algorithms.
>      - **Negation Invariance** provides guidance for tasks that involve flexible relation expressions.
>
> 2. **Practical Applications and Impact** (Sections 4 and 5):
>   - We leverage these metrics in our novel data augmentation framework to **quantify the change of logical consistency** in LLMs. Notably, our experiments demonstrate that the framework enhances logical consistency without compromising alignment with human preferences.
>   - In Section 5, we present a concrete downstream application to highlight the broader implications of logical consistency. The example illustrates **how logical consistency impacts the efficiency and performance of high-level downstream algorithms** in a completely orthogonal way to accuracy.
>
> By addressing logical consistency in both interpretative and practical contexts, **our work lays a foundation for future advancements in logic-dependent algorithm designs and model selection processes**. We hope this clarification helps underscore the broad applicability and value of our findings.

---

> ### Author Response · Authors · 2024-11-22
> **Response to the reviewer (2)**
>
> ---
>
> > W2: The evaluation sets are small, and it is not sure if the findings will be applicable in general.
>
> - We would like to emphasize that in Section 3, we conducted experiments on three diverse and well-established datasets: SummEval, NovelEval, and CaTeRS. These datasets encompass a **wide range of topics** (summarization, document reranking, temporal/causal event ranking) and **levels of subjectiveness**, ensuring that our findings are applicable across various tasks. This broad applicability supports the generalizability of our proposed metrics.
>
> - We also contend that in the era of LLMs, the **diversity** and **quality** of test sets often outweigh sheer quantity. Many recent, highly-regarded benchmarks, such as HumanEval [1] (164 instances) and LLMBar [2] (419 pairwise annotations), are not particularly extensive in size. In contrast, **our datasets include substantially larger pairwise comparison scales**:
>
>     - SummEval: 100 source texts × 16 × 15 (16 summaries for each text) = 24K pairwise comparisons,
>     - NovelEval: 21 queries × 20 × 19 (20 documents for each query) = 8K pairwise comparisons, and
>     - CaTeRS (after filtering): 70 instances × 6.2 × 5.2 (on average 6.2 events for each instance) = 2.3K pairwise comparisons.
>
> ---
>
> > W3: The datasets also come with "candidates" to select. That means LLMs only need to select an answer from candidates. It is not clear if this connects to the concept of logical consistency.
>
> - We acknowledge that our evaluation approach involves selecting answers from given candidates. However, **this paradigm is a common and effective practice for assessing LLMs’ internal understanding**, as evidenced by its widespread use in popular benchmarks such as MMLU [1], ARC [2], and StrategyQA [3].
>
> - In our study, we specifically evaluate LLMs’ judgments to probe their logical consistency based on the fundamental rules of transitivity, commutativity, and negation invariance. **While these rules may not encapsulate the entirety of human logical reasoning, they are essential and widely recognized tools for assessing logical consistency in systems**.
>
> - Additionally, our approach aligns with existing literature, as noted in `Section 6` and expanded upon in `Appendix D`, where we provide further theoretical justification for the importance of transitivity and other logical properties. **Many previous works  [4,5,6]  in NLP and LLM have established that such evaluations are robust indicators of logical consistency**.
>
> In conclusion, **we assert that probing LLMs’ logical consistency through their judgments on well-defined logical properties is both a sound and commonly accepted methodology**. This approach ensures that our evaluation connects meaningfully to the concept of logical consistency.
>
> ---
>
> **References**
>
> [1] Chen, M., Tworek, J., Jun, H., Yuan, Q., Pinto, H. P. D. O., Kaplan, J., ... & Zaremba, W. (2021). Evaluating large language models trained on code. arXiv preprint arXiv:2107.03374.
>
> [2] Zeng, Z., Yu, J., Gao, T., Meng, Y., Goyal, T., & Chen, D. Evaluating Large Language Models at Evaluating Instruction Following. In The Twelfth International Conference on Learning Representations.
>
> [3] Hendrycks, D., Burns, C., Basart, S., Zou, A., Mazeika, M., Song, D., & Steinhardt, J. Measuring Massive Multitask Language Understanding. In International Conference on Learning Representations.
>
> [4] Clark, P., Cowhey, I., Etzioni, O., Khot, T., Sabharwal, A., Schoenick, C., & Tafjord, O. Think you have Solved Question Answering? Try ARC, the AI2 Reasoning Challenge.
>
> [5] Geva, M., Khashabi, D., Segal, E., Khot, T., Roth, D., & Berant, J. (2021). Did aristotle use a laptop? a question answering benchmark with implicit reasoning strategies. Transactions of the Association for Computational Linguistics, 9, 346-361.
>
> [6] Elazar, Y., Kassner, N., Ravfogel, S., Ravichander, A., Hovy, E., Schütze, H., & Goldberg, Y. (2021). Measuring and Improving Consistency in Pretrained Language Models. Transactions of the Association for Computational Linguistics, 9, 1012-1031.
>
> [7] Jang, M., Kwon, D. S., & Lukasiewicz, T. (2022, October). BECEL: Benchmark for consistency evaluation of language models. In Proceedings of the 29th International Conference on Computational Linguistics (pp. 3680-3696).
>
> [8] Asai, A., & Hajishirzi, H. (2020, July). Logic-Guided Data Augmentation and Regularization for Consistent Question Answering. In Proceedings of the 58th Annual Meeting of the Association for Computational Linguistics (pp. 5642-5650).

---

> ### Author Response · Authors · 2024-11-22
> **Response to the reviewer (3)**
>
> ---
>
> > Q1(a): The measures depend on the sampling. How do the scores vary?
>
> We would like to thank the reviewer for bringing this concern up. We believe the reviewer is referring to the calculation of transitivity Equ.1, where we sample M sub-graphs from the original relation graph to estimate the transitivity. We would like to claim that **setting M=1000 (Line 190) should give statistically accurate and stable estimation**. We have included a formal justification in `Appendix H`.
>
> ---
>
> > Q1(b): What are the items (xi or xj) that we put in the equations?
>
> We would like to clarify that ($x_i$, $x_j$) refers to the pairwise comparison by LLMs. **As discussed in Line 110-124, the LLMs are used to produce judgements between any two items with in the item set $X$**, and the items could be different things depends on the tasks. For SummEval, the items are summary candidates, for NovelEval, the items are retrieved documents, and for CaTeRS, the items are event. The prompt templates used to compare items were included in Figure 7.
>
> ---
>
> > Q2: “Are the experiments only conducted in multi-choice datasets that require LLMs to rank the candidates available?”
>
> - We would like to clarify that none of the datasets used in our experiments are classified as "multi-choice" datasets. Instead, each dataset involves multiple items per datapoint with varying characteristics, and the **LLM judgments are produced through pairwise comparisons**. We did not explicitly ask the LLMs to rank the candidates, instead we evaluated the consistency of LLMs based on all the pairwise judgements.
>
> - **This approach does not limit the generalizability of our framework.** Our methodology is applicable across a wide range of domains, as discussed in detail in `W2` and `W3`, where we elaborate on the diversity of topics and the relevance of the selection-based evaluation format.
>
> ---
>
> > Q3: “The paper doesn't show the details about the instruction-tuning settings.”
>
> - We would like to emphasize that **all training configurations** for the instruction-tuning experiments are detailed in `Appendix E` (as referred by line 453). Additionally, **information about the Summary with Feedback** dataset used in these experiments is provided in `Appendix B` (as referred by line 446).
>
> - To address the reviewer’s concerns further, we have now **included the exact instruction prompt templates** utilized for instruction-tuning in `Appendix F`. We hope this addition ensures full transparency and reproducibility of our methodology.
>
> ---
>
> Once again, we thank the reviewer for their time and detailed feedback. If the reviewer has any further questions or suggestions, we would be more than happy to address them. We hope that, in light of our overall contributions and the responses provided, the reviewer might consider reevaluating the score.

---

> ### Comment · Reviewer_rqBr · 2024-11-25
> **logical consistency vs. partial order of preference ranking**
>
> Logical consistency is not defined in this paper.
> The referred papers [4, 5, 6] also don't define it.
> In the rebuttal, it was mentioned that many previous works [4,5,6] in NLP and LLM have established that such evaluations are robust indicators of logical consistency.
> However, I couldn't find such statements.
> [4, 6] don't mention logic anywhere.
> [5] mentions logic when it mentions a comparator operator.
>
>
> [1, 2, 3] are datasets for the natural language tasks.
> After rephrasing the format as question-answering and limiting it to multiple-choice questions, we have a universal task that covers various tasks.
> This doesn't imply that "it is a common and effective practice for assessing LLMs’ internal understanding."
> Indeed, none of the three papers claimed anything about the internal understanding of LLMs.
>
>
> Without a definition of logical consistency, it is unclear how to quantify undefined concepts.
> The implicit assumption in this paper is having high scores on the three evaluation metrics implies higher logical consistency.
> It is unclear how such an implication is true.
> It is unclear how to use this for logic-dependent algorithm design and model selection.
> It is unclear what a logic-dependent algorithm is.
>
> From what's written, what's measured in this paper is the preference or the order of items that should satisfy a few axioms.
> When a preference is combined with logic, axioms could be introduced into the knowledge base.
> For example, such a KB can be seen in ["A logical reasoning with preference," S.K. Das, 1995].
>
> Most places that mention "logical" in this paper refer to the partial order.
> When this paper claims "logical consistency," it is about consistency with respect to the partial order extracted from LLM over the items.
> In other words, claims about " logical consistency evaluation " need to be carefully examined to see if they can extend beyond checking the order of preference.
>
> The relationship examined in this paper is the preference/order between two items.
> The function F could be seen as a predicate such as Preferred(x, y), with x and y being any objects in the knowledge base.
> The predicate means that x is preferred to y.
> The paper's main contribution is designing F as prompts for LLM, shown in Appendix F.
> Figures 7 and 8 show that the predicate is written in natural language sentences, and the objects are two outputs from LLM for a certain task.
>
> It would be more precise to say "LLM as a natural language prompt of a binary preference relation" than "LLM as logical operators."
> This paper would be closer to measuring/improving the prediction of LLM on the ranking of multiple candidates from various NLP tasks.
>
> It is unclear that "this framework broadly applies to any task where logical relations are relevant."
> If that is the case, the authors should prove or empirically show it.

---

> > ### Author Response · Authors · 2024-11-27
> > **Response to the reviewer Round 2 (2)**
> >
> > **Concern 3: Definition of Logical Consistency and Metrics**
> >
> > > Without a definition of logical consistency, it is unclear how to quantify undefined concepts. The implicit assumption in this paper is having high scores on the three evaluation metrics implies higher logical consistency. It is unclear how such an implication is true.
> >
> > We respectfully disagree with the assessment that the paper lacks a clear definition of what is being measured. **We explicitly define the scope of logical consistency we address in the paper**:
> >
> > - Line 080: “We focus on three fundamental aspects of logical consistency: transitivity, commutativity, and negation invariance.”
> > - Line 110-116: “We evaluate the logical consistency of LLMs by assessing their ability to predict logically consistent relations among a set of items. These items could represent diverse entities or events with a uniform relation between them; such a relation might be (i) comparing the preference among response candidates to a query or (ii) determining the causal order of shuffled events, among other possibilities. This evaluation is grounded in relational algebra and order theory (Abiteboul et al., 1995; Imielinski & ´ Lipski Jr, 1984), with the goal of assessing whether the model maintains coherent and self-consistent judgments across its predictions.”
> >
> > **While we recognize that the reviewer may hold a different perspective on whether these aspects should belong to logical consistency, we believe it is evident that our scope and definitions are clearly articulated.**
> >
> > Regarding the relationship between the evaluation metrics and logical consistency, we note that the three metrics directly quantify the degree to which the logical consistency rules (transitivity, commutativity, and negation invariance) are satisfied. These rules are rooted in relational algebra and order theory, as cited in the paper, and serve as the foundation for our quantification methodology.
> >
> > ---
> >
> > **Concern 4: Usefulness for Logic-Dependent Algorithm Design and Model Selection**
> >
> > > It is unclear how to use this for logic-dependent algorithm design and model selection. It is unclear what a logic-dependent algorithm is.
> >
> > **We believe the paper provides a clear definition of logic-dependent algorithms.** We elaborate further in several parts of the paper:
> >
> > - Line 097: "They show better performance and reduced computational requirements when employed as logical operators in a sorting-based preference ranking aggregation algorithm (Liu et al., 2024) that relies on logical coherence."
> >
> >
> > - Line 467-474: "We explicitly show the examples of what we mean by high-level algorithms, including document reranking for retrieval systems, optimization in prompt-tuning algorithms, and pairwise comparators for rank aggregations."
> >
> > **The paper also demonstrates how logical consistency performance directly impacts algorithmic outcomes, and provide the guidance for the model selection**:
> >
> > - Line 472: "When LLMs are used as logical operators, maintaining a high level of logical consistency is critical to ensure predictable and efficient decision-making."
> >
> > - Line 490-495: "Our findings from Table 4 reveal two key insights: 1) Although Phi-3-mini has slightly lower accuracy with human judgments than GPT-3.5-turbo, its stronger transitivity leads to better ranking performance with or without calibration. 2) There is a clear correlation between an LLM’s commutativity and the performance gains from calibration in the PairS algorithm. LLMs that are more commutative rely less on calibration to achieve optimal performance, and therefore, require less computations."
> >
> > **Additionally, we explicitly outline guidance for designing algorithms based on consistency performance.** For example:
> > - “When LLMs generally exhibit low logical consistency in a certain domain or task, the high-level algorithm should be designed with less reliance on logic. For instance, in the task of rank aggregation, a sorting-based algorithm might be unsuitable, and a more robust alternative such as ELO rating may be preferable.”
> >
> > ---

---

> > > ### Comment · Reviewer_rqBr · 2024-11-27
> > >
> > > In general, logic consistency refers to the set of statements that don't contradict each other.
> > >
> > > When you say logic consistency, it is restricted to the consistency over the knowledge base where it has axioms for transitivity, negation, etc as shown in the example paper that introduces the preference to the logic.
> > > Those three rules can be introduced to capture the preference, but it wouldn't be fair to say that they are fundamental aspects of logic consistency as we could design any knowledge base that is equipped with different rules.
> > >
> > > As mentioned earlier, when this paper mentions "logic consistency," it can be replaced by "consistency with respect to the partial order of preference statements", which has a very specific scope. I am not saying that being specific on the scope will hurt the value of the work.

---

> ### Comment · Reviewer_rqBr · 2024-11-25
> **instruction tuning and its impact to preference ranking task**
>
> Keeping aside the issues around imprecise description of the work around logic, the content of the paper shows empirical results around the predictive behavior of various LLMs.
> Although there is a gap in stating that such findings will lead to the internal understanding of LLM around logical reasoning, they still produce useful information. It depends on the interpretation, but the main takeaway would be summarized in Table 1 over K=5.
>
>
> The next main contribution of this paper would be extracting fine-tuning data from the noisy partial order and aligning the LLM behavior to improve the consistency on the preference rankings. This also connects to the impact on the performance of ranking task in Table 4, where the results indicate that the higher consistency model performs better.
>
> Question: Is this improvement still observed by improving the consistency of a fixed model? Since Table 4 shows five different models, it would be clear if we see that the improved model after fine-tuning in Section 4 indeed shows the improved performance in the Table 4 setting.

---

> ### Author Response · Authors · 2024-11-27
> **Response to the reviewer Round 2 (1)**
>
> We sincerely thank the reviewer for their detailed and thoughtful feedback. We greatly appreciate the time and effort you have devoted to evaluating our work. We address the concerns raised below:
>
> ---
> **Concerns 1: Definition of Logical Consistency**
>
> > Logical consistency is not defined in this paper. The referred papers [4, 5, 6] also don't define it. In the rebuttal, it was mentioned that many previous works [4,5,6] in NLP and LLM have established that such evaluations are robust indicators of logical consistency. However, I couldn't find such statements. [4, 6] don't mention logic anywhere. [5] mentions logic when it mentions a comparator operator.
>
> We apologize for the oversight in our previous rebuttal, where references [4, 5, 6] were mistakenly cited. The correct references should have been [6, 7, 8]. **We would like to emphasize that defining transitivity, commutativity, and negation invariance as rules for determining logical consistency is a well-established approach in prior works.** These rules assess the logical consistency of language models from certain perspective, and, as mentioned in previous rebuttal, we have extensively elaborated on this connection in Section 6 and Appendix D of our paper. To further clarify, we explicitly list below the relevant prior works that align with this view:
>
> - *BECEL: Benchmark for Consistency Evaluation of Language Models*:
>
>   This work discusses **negational consistency, symmetric consistency** (equivalent to our commutativity consistency), **transitive consistency**, and additive consistency as forms of **logical consistency** in Section 2.
>
> - *Logic-Guided Data Augmentation and Regularization for Consistent Question Answering*:
>
>   **Logical rules** for consistent QA are covered in Section 3.1, including **symmetric and transitive consistency**.
>
> - *A Logic-Driven Framework for Consistency of Neural Models*:
>
>   Annotation consistency, **symmetry consistency, and transitivity consistency** are analyzed as **logical consistency** in Section 3.1.
>
> - *Consistency analysis of chatgpt.*:
>
>   Semantic consistency, **negation consistency, symmetric consistency, and transitive consistency** are discussed as **logical consistency** in NLI tasks in Section 1.
>
> - *Adversarially Regularising Neural NLI Models to Integrate Logical Background Knowledge*:
>
>   **Symmetric**, reflexive, and **transitive rules** are presented as structural constraints for human **reasoning** in Section 1.
>
> Additionally, on the theoretical side:
>
> - *Intransitivity of preferences*:
>
>   They explored psychological factors leading to **intransitive** preferences, illustrating deviations from **rational models**.
>
> - *On the calculus of relations*:
>
>   They examined **transitivity** in relational calculus, foundational to many **logical systems**.
>
> - *The structure of values and norms*:
>
>   They provided insights into the implications of violating **transitivity** in preferences, showing that such violations can lead to **logical inconsistencies** within decision theory.
>
> - *The sorites paradox*:
>
>   They proved that apparent violations of **transitivity** can give rise to **logical paradoxes**
>
> **These works collectively establish the theoretical and practical basis for our approach to defining logical consistency.** The detailed citation can be found in the paper.
>
>
> ---
>
> **Concern 2: Definition of Internal Understanding**
>
> > [1, 2, 3] are datasets for the natural language tasks. After rephrasing the format as question-answering and limiting it to multiple-choice questions, we have a universal task that covers various tasks. This doesn't imply that "it is a common and effective practice for assessing LLMs’ internal understanding." Indeed, none of the three papers claimed anything about the internal understanding of LLMs.
>
> We respectfully disagree with this assessment. The MMLU, ARC, and StrategyQA datasets are widely recognized as benchmarks for evaluating a language model’s internal knowledge and understanding. **These datasets require language models to generate answers based solely on their own knowledge and reasoning, without the aid of external information.** This evaluation inherently reflects the model’s internal understanding. We would greatly appreciate it **if the reviewer could clarify their interpretation of "internal understanding" and provide any specific definitions or references that support their assessment.** This would help us better address the concerns raised and refine our arguments accordingly.
>
> To clarify, what we are arguing in previous rebuttal is that **the format of multiple-choice tasks effectively captures a model’s internal understanding.** Therefore, **our setup, which involves LLMs making binary judgments, is a natural adaptation for assessing the consistency of their understanding.** We believe this argument is well-supported by both the structure of these datasets and their widespread use in evaluating LLMs.

---

> > ### Comment · Reviewer_rqBr · 2024-11-27
> >
> > **Logical consistency**
> >
> > * [6] is on the consistency of a model for predicting the same answer for two paraphrases.
> >
> > * [7] is close to the submitted paper.
> > It defines consistency metrics similar to the proposed metrics. This paper needs to be referred to more directly than put inside as a motivation or related work. [7] defines similar metrics reflecting the logical properties, but its usage and propositions are different. The logical properties are introduced with logical statements, and the scope of the work is clearly described.
> > When I see [7], the difference is only the models tested. The overall framework is the same. The contribution of the submission is to extend the earlier work to large language models with prompting to measure the consistency metrics and propose a fine-tuning method that improves the consistency metrics.
> >
> > Overall, the issue is the overstatements in the paper and the rebuttals.
> >
> > * [8] also shows an approach that uses symmetry and transitive metrics to generate consistent answers for QA tasks. The models are not LLMs, but the overall idea is the same. It also shows data augmentation schemes.
> >
> > The authors stated in the abstract that this is the first universal framework to quantify logical consistency. However, this contradicts the statement in the rebuttal, saying that rules for determining logical consistency is a well-established approach in prior works. I don't mind whether one method comes first or not. One came after can extend or solve problems that didn't exist in the past. The current work is tailored to LLMs and comes with prompts.
> >
> >
> > **Definition of internal understanding**
> >
> > The definition of internal understanding of LLM would be understanding the mechanisms of LLMs so that such findings on the internal understanding explain how the output was generated given some input.
> >
> > From what's written ("...these datasets require language models to generate answers based solely on their own knowledge and reasoning..."), I think it is a problematic statement.
> > It assumes that LLMs generate answers by reasoning over their knowledge by treating a machine learning model like a person. It is not clear if LLMs are performing reasoning at this point.
> >
> > While seeing "...the format of multiple-choice tasks effectively captures a model’s internal understanding",  I think this is also problematic since it is not a verified statement.
> > It seems that the authors use the meaning of internal understanding as more like the behavior of the model.

---

> > > ### Author Response · Authors · 2024-11-28
> > > **Response to the reviewer Round 3 (1)**
> > >
> > > We appreciate the reviewer for the further clarification and feedbacks. We address the concerns below:
> > >
> > > ---
> > >
> > > **Scope of our work**
> > >
> > > > The authors stated in the abstract that this is the first universal framework to quantify logical consistency. However, this contradicts the statement in the rebuttal, saying that rules for determining logical consistency is a well-established approach in prior works.
> > >
> > > **We respectfully request the reviewer to carefully revisit our claims.** We are **not** asserting that ours is "the `first` universal framework to quantify logical consistency."** As stated in the abstract: “We `first` propose a universal framework to quantify logical consistency via three fundamental proxies: transitivity, commutativity, and negation invariance.” Our claim is specifically scoped and does not imply exclusivity as the "first" in the field.
> > >
> > > > [7] is close to the submitted paper. It defines consistency metrics similar to the proposed metrics. This paper needs to be referred to more directly than put inside as a motivation or related work.
> > >
> > >
> > > To address the reviewer’s concerns, we are willing to **move the discussion of prior work on transitivity, negation, and commutativity to the introduction** for better clarity. However, we emphasize again that prior work has largely been constrained to analyzing consistency **between two or three statements** within limited domains such as **NLI and QA**. These approaches typically identify **specific contradictory patterns** among two or three statements.
> > >
> > > In contrast, our framework generalizes to **any number of items or statements** and introduces a **novel** method to quantify transitivity. Furthermore, our framework extends beyond domain-specific applications to **general domains equipped with logical consistent relations**. This is the basis for our claim of universality within the specified scope, which is clearly stated in the paper (lines 515–520) and reiterated in this rebuttal. If the reviewer identifies this as a primary issue, we are happy to revise and bring this discussion forward to the introduction section.
> > >
> > > ---
> > >
> > > **Definition of internal understanding**
> > >
> > > > The definition of internal understanding of LLM would be understanding the mechanisms of LLMs so that such findings on the internal understanding explain how the output was generated given some input.
> > >
> > >
> > > We understand the reviewer’s perspective on defining the internal understanding of LLMs as "understanding the mechanisms of LLMs such that findings explain how outputs are generated given inputs." However, we believe this pertains to **understanding the internal mechanisms of LLMs**, which differs from the concept of **LLM’s internal understanding**.
> > >
> > > That said, we acknowledge that terminological interpretations may vary. If rephrasing *“LLM’s internal understanding”* to *“LLM’s internal judgments”* would reduce ambiguity and address the reviewer’s concern, we are more than willing to make this adjustment.
> > >
> > > ---

---

> ### Author Response · Authors · 2024-11-27
> **Response to the reviewer Round 2 (3)**
>
> **Concern 5: Scope of Relations and Logical Consistency in Rankings**
>
> > From what's written, what's measured in this paper is the preference or the order of items that should satisfy a few axioms. When a preference is combined with logic, axioms could be introduced into the knowledge base. For example, such a KB can be seen in ["A logical reasoning with preference," S.K. Das, 1995].
>
> We appreciate the reviewer’s suggestion regarding preference axioms. However, **we would like to emphasize that our scope is to extend the logical consistency evaluation of transitivity, commutativity, and negation invariance from local 2-3 items to arbitrary number of items.** This inherently leads to the construction of full or partial rankings when logical relations are coherent.
>
> Previous works have defined these logical consistency rules as constraints in NLI and QA. Our contribution lies in generalizing them to multiple items and general domains with logical relations, and developing metrics that quantify the degree of consistency achieved. Thus, **the rankings emerge as a consequence of applying logical consistency rules, rather than as an input assumption.**
>
> ---
>
> **Concern 6: Definition of Logical Operator**
>
> > It would be more precise to say "LLM as a natural language prompt of a binary preference relation" than "LLM as logical operators." This paper would be closer to measuring/improving the prediction of LLM on the ranking of multiple candidates from various NLP tasks.
>
> We appreciate the reviewer’s comment and would like to address the suggestion. We are somewhat unclear about the reasoning behind the statement. In the context of a sorting algorithm, the symbol ‘>’ is commonly understood as a **logical operator**, as it is expected to make logically consistent judgments when comparing two numbers.
>
> Similarly, when using LLMs within logic-dependent high-level algorithms, we expect them to exhibit logical consistency when making pairwise judgments between items. This aligns with the behavior typically associated with logical operators.
>
> **If the reviewer has a different interpretation of whether ‘>’ qualifies as a logical operator, we are open to rephrasing it as a "logical comparator" to ensure clarity and alignment with their perspective.**
>
> ---
>
> **General Response**:
>
> We sincerely appreciate the reviewer’s thoughtful feedback. **It seems there may be differences in how logical consistency is understood and defined**. We acknowledge that as the field of NLP evolves rapidly, variations in terminology and interpretations can arise. However, **we want to emphasize that our work builds upon and aligns with established understandings in prior research, which clearly consider transitivity, commutativity (or symmetry), and negation invariance as fundamental logical consistency rules**.
>
> **We kindly request the reviewer to consider the broader contributions of our work, beyond potential differences in definitions, as we believe our findings and methodologies offer meaningful advancements in evaluating and improving consistency in LLMs, and also demonstrating their impacts.**

---

> > ### Comment · Reviewer_rqBr · 2024-11-27
> >
> > I think there's a conceptual misunderstanding. When we say a symbol '>' in logic, you could define a predicate such as GreaterThan(X, Y) defined over two objects X and Y, and give a semantic that the predicate serves as a comparator operator.
> > It doesn't sound good to hear that it is a logical operator or logical comparator.
> > Although the field of NLP evolves rapidly, such a basic definition wouldn't change.

---

> ### Author Response · Authors · 2024-11-27
> **Response to the reviewer Round 2 (Question)**
>
> **Additional Experiments**
>
> > Question: Is this improvement still observed by improving the consistency of a fixed model? Since Table 4 shows five different models, it would be clear if we see that the improved model after fine-tuning in Section 4 indeed shows the improved performance in the Table 4 setting.
>
> We thank the reviewer for suggesting this experimental setup, which indeed provides a **fair comparison**. However, with the current *Summary with Feedback dataset*, this specific experiment is **not directly feasible**. The dataset contains sparse pairwise annotations, and there is **no human-provided ranking** against which the estimated rankings from PairS can be compared.
>
> To address the reviewer’s concern, we conducted **additional instruction-tuning experiments** using a different **document reranking** dataset, *MS MARCO* (details provided in Appendix B). This dataset allows for a more robust evaluation while maintaining consistency with the experimental setup of Table 3 in the manuscript. We also perturbed 10% of the pairwise comparison labels to test the robustness of the models.
>
> Below, we first present the consistency performance results (equivalent to Table 3) for the fine-tuned model...
>
> | Models                   | Human Acc. | $S_{tran}(5)$ | $S_{comm}$ | $S_{neg}$ |
> |--------------------------|:----------:|:--------:|:-----:|------|
> | Zero-shot inference      |    57.2    |   74.2   |  68.5 | 64.7 |
> | IT perturbed data        |    74.7    |   86.7   |  81.1 | 62.9 |
> | IT augmented data        |    75.0    |   91.3   |  85.9 | 63.1 |
> | IT augmented data + Neg. |    74.9    |   90.2   |  86.2 | 85.5 |
>
> The results reaffirm the effectiveness of our proposed data augmentation framework, where the consistency is improved and the alignment to human preference is maintained.
> Then we show that the **Spearman correlations** of the predicted rankings by PairS algorithm.
>
>
> | Model             | Human | $S_{tran}(5)$ | $S_{comm}$ | PairS | PairS calibrated |
> |-------------------|:-----:|:--------:|:-----:|:-----:|:----------------:|
> | zero-shot         |  57.2 |   74.2   |  68.5 |  18.7 |       22.1  (+3.4)     |
> | IT perturbed data |  74.7 |   86.7   |  81.1 |  58.1 |       60.3  (+2.2)     |
> | IT augmented data |  75.0 |   91.3   |  85.9 |  61.0 |       62.6   (+1.6)    |
>
>
> The results also agree with our findings in Section 5, where models with **better transitivity can produce overall more accurate rankings**, and models with **better commutativity rely less on calibration to achieve optimal performance**.

---

> > ### Comment · Reviewer_rqBr · 2024-11-27
> >
> > This is about Section 5.1, which discusses the impact of consistency on the PairS applications.
> >
> > Table 4 shows the results over five different models and states that "improving the metric" implies "improved performance."
> > When we see the table with five models, there's no clear or strong trend.
> > Phi-3-med is the best in both transitivity and PairS.
> > Mistral is the second best in transitivity but the worst in PairS.
> > Phi-mini is better than Llama-3 and GPT-3.5 in transitivity, but it is worse than Llama-3 in PairS.
> >
> > One unverified statement is that "fine-tuning" the data augmentation scheme proposed in the paper will improve the performance of the downstream task. It would support the statement in Section 5.1 if that could be shown.

---

> > > ### Author Response · Authors · 2024-12-02
> > > **Reminder to the reviewer**
> > >
> > > Thank you for taking the time to review our paper, provide thoughtful feedback, and actively participate in the discussion. As the discussion period is drawing to a close, we kindly ask if our previous rebuttal and additional experiments have addressed your concerns. If there are any remaining issues or points of clarification, we are more than willing to provide further details.
> > >
> > > Additionally, we respectfully encourage the reviewer to consider the overall contributions of this work and to revisit the score in light of our responses. Your insights and evaluation are greatly appreciated.

---

> ### Author Response · Authors · 2024-11-28
> **Response to the Reviewer Round 3 (2)**
>
> **Definition of Logical Consistency**
>
> > In general, logic consistency refers to the set of statements that don't contradict each other.
>
> We appreciate the reviewer’s definition that “logical consistency refers to the set of statements that don't contradict each other,” and we fully agree with this interpretation. Within this broad framework, **we argue that transitivity, commutativity, and negation are indeed encompassed as logical consistency rules**.
>
> - For example, statements such as “A is better than B” and “B is better than A” clearly contradict each other when evaluated under the same prompt template and criteria.
>
> Similarly, contradictions arising from failures in transitivity and negation demonstrate their relevance to logical consistency.
>
>
>
> > When you say logic consistency, it is restricted to the consistency over the knowledge base where it has axioms for transitivity, negation, etc as shown in the example paper that introduces the preference to the logic. Those three rules can be introduced to capture the preference, but it wouldn't be fair to say that they are fundamental aspects of logic consistency as we could design any knowledge base that is equipped with different rules.
>
> When the axioms applied to a knowledge base or a set of items are explicitly designed to prevent **logical contradictions**, we believe it is reasonable to describe this as logical consistency. While logical consistency is indeed a broad term, our work focuses on specific aspects of it, with a well-defined scope centered on transitivity, commutativity, and negation.
>
> If the reviewer finds our use of the term *“fundamental aspects of logical consistency”* too strong or potentially misleading, we are happy to revise it to a softer phrasing such as *“particular/arithmetic aspects of logical consistency.”* This adjustment would ensure a clear emphasis that our focus is on certain components of logical consistency, rather than its entirety.
>
>
> > As mentioned earlier, when this paper mentions "logic consistency," it can be replaced by "consistency with respect to the partial order of preference statements", which has a very specific scope. I am not saying that being specific on the scope will hurt the value of the work.
>
> We also acknowledge the reviewer’s point regarding the specific scope of “consistency with respect to the partial order of preference statements” when the concepts of transitivity, negation and commutativity are extended to a set of items. However, we would like to highlight that **partial order preference consistency is also widely recognized as a type of logical consistency**, particularly in fields such as decision theory, preference modeling, and social choice theory. **Our terminology aligns with the understanding established in prior works** (listed in previous rebuttal), and we have made our scope explicitly clear from the outset of the paper.
>
> ---
>
> **Definition of Logical operator**
>
> > I think there's a conceptual misunderstanding. When we say a symbol '>' in logic, you could define a predicate such as GreaterThan(X, Y) defined over two objects X and Y, and give a semantic that the predicate serves as a comparator operator. It doesn't sound good to hear that it is a logical operator or logical comparator. Although the field of NLP evolves rapidly, such a basic definition wouldn't change.
>
> We agree there may be a conceptual misunderstanding here. The reviewer seems to be interpreting the symbol ">" in the context of **narrow-sense formal or predicate logic**, while our usage refers more broadly to **arithmetic or quantitative logic**.
> - The arithmetic logic deals with numerical values and their relationships, typically involving mathematical operations like addition, subtraction, multiplication, and comparison (e.g., greater than, less than). It is a specific subset of logic that deals with quantities and the logical relationships between them.
>
> If this is the root of the confusion, we are happy to clarify and specify the term as "arithmetic logical operator" to avoid ambiguity, and also explicitly mention this in the beginning of the paper.

---

> ### Author Response · Authors · 2024-11-28
> **Response to the reviewer Round 3 (3)**
>
> > This is about Section 5.1, which discusses the impact of consistency on the PairS applications.
>
> > Table 4 shows the results over five different models and states that "improving the metric" implies "improved performance." When we see the table with five models, there's no clear or strong trend. Phi-3-med is the best in both transitivity and PairS. Mistral is the second best in transitivity but the worst in PairS. Phi-mini is better than Llama-3 and GPT-3.5 in transitivity, but it is worse than Llama-3 in PairS.
>
> > One unverified statement is that "fine-tuning" the data augmentation scheme proposed in the paper will improve the performance of the downstream task. It would support the statement in Section 5.1 if that could be shown.
>
> **We respectfully request that the reviewer revisit our [rebuttal](https://openreview.net/forum?id=kJgi5ykK3t&noteId=qrPOaRe2AF
> )**, as we have indeed addressed this point. Specifically, we demonstrated that the model fine-tuned on augmented data outperforms the model fine-tuned on un-augmented perturbed data in the context of the PairS algorithm.
>
> ---
>
> We hope this clarification addresses the reviewer’s concerns, and we remain open to further refining our terminology or phrasing to ensure clarity and accuracy.

---

### Official Review · Reviewer_xkx3 · 2024-11-01

**Soundness:** 2
**Presentation:** 2
**Contribution:** 2
**Rating:** 6
**Confidence:** 4

**Summary:**

The paper studies how to improve logical consistency in Large Language Models (LLMs) to make them more reliable for decision-making tasks. It introduces a framework to measure logical consistency using three key properties: transitivity (keeping decisions consistent over a series of comparisons), commutativity (order of comparison doesn’t affect results), and negation invariance (consistent handling of opposite statements). A technique for data refinement and augmentation is proposed to improve the logical consistency and then improve the performance in logic-dependent algorithms.

**Strengths:**

- Logical consistency is an interesting issue in LLMs, such as handling cases where both A > B and A < B appear; how should the final ranking be determined?
- Clearly explain the main idea in Fig. 1.
- Three types of inconsistency are proposed: transitivity (e.g., if A > B and B > C, but A < C), commutativity (e.g., A > B and A < B), and negation.

**Weaknesses:**

- Win-loss rate preference ranking is one way to solve the noisy pairwise annotations. It would be interesting to explore how other models, such as Borda rule and others from social choice theory—where rankings often conflict—might help with this issue.
- There is a lack of analysis on the extent to which the proposed technique, win-loss rate preference ranking, is effective and when it may fail.

**Questions:**

- Could you provide more details on situations where the win-loss rate preference ranking might fail? What specific limitations did you observe?
- Could you offer more theoretical analysis on why the proposed win-loss rate preference ranking improves LLM consistency? LLMs might sometimes guess the correct answer without understanding it.
- Is it possible to have LLMs analyze why each step or change in the proposed technique works, to ensure they operate in the intended way?
- Have you compared the win-loss rate approach to other ranking methods, such as the Borda rule or similar techniques? If so, what differences did you find in handling noisy data?

---

> ### Author Response · Authors · 2024-11-20
> **Response to the reviewer (1)**
>
> We sincerely thank the reviewer for their time and effort in reviewing our paper. We greatly appreciate their insightful suggestions and believe these will significantly enhance the quality of our work. Below, we address and clarify the concerns raised:
>
> ---
>
>
> > W1 & Q4: Compare the win-loss rate rank estimation with other ranking methods.
>
> We are grateful to the reviewer for highlighting this important point. As mentioned in the first paragraph of Section 4.1, we acknowledge that alternative rank estimation methods might yield better performance. Our choice of the win-loss rate is intended as a proof-of-concept, and we do not claim it achieves the best possible performance.
>
>
> To address the reviewer’s concerns more thoroughly, **we conducted additional experiments comparing win-loss rate with other ranking methods**.
> While the Borda count is a commonly used ranking method, it is not directly applicable to our setting. Borda is typically designed for scenarios involving complete or partial ranking votes, whereas our dataset contains only pairwise annotations. In this context, the Borda count effectively reduces to win-loss count. However, win-loss count does not normalize for the number of comparisons per item, leading to potential unfairness when preference annotations are imbalanced (e.g., some items are compared more frequently than others). This motivated our choice of the win-loss rate, which accounts for this imbalance.
>
> For a more comprehensive comparison, we also evaluated two widely-used ranking methods: the ELO rating system and the Bradley-Terry (B-T) model. Below, we present results from experiments using these methods to refine the data, with all other experimental setups remaining consistent with Table 3 in the manuscript.
>
>
> | Models              | Human Acc. | Stran(5) | Scomm |
> |---------------------|:----------:|:--------:|:-----:|
> | Zero-shot inference |    62.9    |   84.7   |  68.2 |
> | IT perturbed data   |    71.3    |   92.4   |  79.6 |
> | IT augmented **W-L**    |    70.5    |   97.5   |  89.4 |
> | IT augmented **ELO**    |    71.4    |   97.7   |  89.7 |
> | IT augmented **B-T**    |    71.2    |   98.6   |  91.0 |
>
>
> *Our findings indicate that both ELO and the B-T model achieve slightly better performance than the win-loss rate, with improvements observed in both human accuracy and consistency metrics. However, the performance margins remain relatively small, further supporting our decision to use win-loss rate as a proof-of-concept in the initial study.*
>
> ---

---

> ### Author Response · Authors · 2024-11-20
> **Response to the reviewer (2)**
>
> ---
>
> > W2 & Q1: Lack of analysis on when win-loss rate rank estimation is effective and when it may fail.
>
>
> We appreciate the reviewer’s observation regarding~ the need for a detailed analysis of when win-loss rate rank estimation is effective and when it may fall short. Below, we provide a comparative analysis of the strengths and weaknesses of different rank estimation methods:
>   - **Win-loss rate** does not incorporate transitive information, meaning comparisons against weaker or stronger candidates are treated with equal weight. While this might seem like a limitation, it offers advantages in specific cases. We show the pros and cons in belows:
>     - Pros: In the example with pairwise comparisons [A>B, A>C], there is insufficient information to infer the relationship between B and C. The win-loss rate can successfully predict A>B=C. We will show later how ELO and B-T fail to do this.
>     - Cons: In the example of [A>B, B>C, C>D], there should be enough information to derive that A>B>C>D. However, the win-loss rate will produce A>B=C>D, as it treats all comparisons with the same importance, regardless of the broader context.
>
>
>
> - **ELO rating** effectively incorporates the transitive information, however, it is highly sensitive to the order of comparisons. When comparison is sparse this will significantly affect the estimated ranking. For example:
>     - In the case [A>B, A>C], ELO will predict A>C>B. However, by reversing the comparison order to [A>C, A>B], ELO will predict A>B>C. Since our dataset is static, the order of comparison contains no inherent information. Being sensitive to order will potentially introduce noise to the augmented preference data.
>
>
>
> - **Bradley-Terry model** produces maximum likelihood estimations for rankings, which offers a theoretical advantage. It effectively handles transitive relationships and provides more nuanced rankings. However, being a parametric model requiring optimization, the B-T model struggles to output tied rankings (cannot predict the same score for two items). In other words, it will always produce complete rankings instead of partial rankings, unless some special design on the tolerance. For example:
>     - In the case [A>B, A>C], the B-T model will produce A>B~C, where B and C have slightly different scores, even if they should theoretically tie. This lack of tolerance for ties will potentially introduce noise into our augmented data.
>
>
>
> In general, the effectiveness of any rank estimation method depends on several factors, including the sparsity of pairwise comparison annotations and the available computational resources. For instance:
>   - Sparse comparisons: Methods that heavily rely on transitive relationships, like Elo and B-T, might struggle when comparison data is limited.
>   - Time constraints: Simpler methods like win-loss rate are computationally efficient and may be preferable when time is a limiting factor.
>
> Overall, while methods such as Elo and the B-T model have specific advantages in capturing transitive relationships or producing likelihood-based rankings, their sensitivity to noise and computational complexity make them less practical under certain conditions. By contrast, the win-loss rate provides a straightforward and robust baseline, particularly when tied rankings are acceptable or when computational simplicity is prioritized.

---

> ### Author Response · Authors · 2024-11-20
> **Response to the reviewer (3)**
>
> ---
> > Q2: Could you offer more theoretical analysis on why the proposed win-loss rate preference ranking improves LLM consistency?
>
>
> As discussed in Lines 413–416 and Lines 422–423, **the primary motivation behind our data augmentation and refinement framework is to generate consistent, conflict-free preference data**. Our hypothesis is that models trained on data with explicit consistency are more likely to exhibit consistent behavior themselves, as they are less exposed to conflicting signals during training. The experimental results validate this hypothesis by demonstrating measurable improvements in logical consistency.
>
>
> We would also like to emphasize that utilizing data augmentation techniques to enhance specific abilities of an LLM is a well-recognized and intuitive approach in the field. By systematically refining training data to align with the intended task objectives, our method ensures the LLM receives clearer signals for learning consistent preferences, thus improving performance in this dimension.
>
> ---
>
>
> > Q3: "Is it possible to have LLMs analyze why each step or change in the proposed technique works, to ensure they operate in the intended way?”
>
> We are **unclear about the specific intent of this question** and would appreciate further clarification from the reviewer. Specifically, it is not clear what is meant by “the proposed technique” and why there is a need for the LLM itself to “analyze” the framework.
> If the question refers to the data augmentation framework, we have already provided a detailed explanation of its workflow and motivations in Lines 370–416 and Figure 6. If the reviewer is seeking additional insights beyond what we have presented, we would be happy to address those concerns with more specifics once clarified.
>
> ---
>
> **General response**:
>
> We recognize that the reviewer’s primary concerns are centered around the win-loss rate rank estimation. While we acknowledge that more sophisticated ranking methods could potentially lead to improvements, we have conducted additional experiments to compare alternative methods and provided those results to address this point.
>
> However, we would like to stress that the contributions of this work extend beyond the choice of ranking method:
>   1. **Development of Logical Consistency Metrics**: We introduced three logical consistency metrics, which provide a novel perspective for evaluating LLMs.
>   2. **Comprehensive Evaluation**: We assessed the performance of a broad range of LLMs across different setups, offering valuable insights into their behavior.
>   3. **Proposed Data Augmentation and Refinement Framework**: We demonstrated that this framework effectively enhances logical consistency when aligning LLM preferences, a contribution validated by our experiments.
>   4. **Demonstrating the Importance and impact of Logical Consistency**: We showed that logical consistency is a critical factor, in addition to alignment with human preferences, for the success of logic-dependent, high-level algorithms.
>
> ---
>
> Once again, we thank the reviewer for their time and detailed feedback. We will revise our manuscript based on the above discussions accordingly. We hope that, in light of our overall contributions and the responses provided, the reviewer might consider reevaluating the score.

---

### Official Review · Reviewer_4Hhg · 2024-11-02

**Soundness:** 4
**Presentation:** 4
**Contribution:** 3
**Rating:** 6
**Confidence:** 4

**Summary:**

This paper introduces a comprehensive framework that measures logical consistency using three different metrics. The study evaluates these consistency measurements systematically, using the findings as indicators of overall model robustness. It also demonstrates a proof-of-concept method for improving LLM consistency through data refinement and augmentation. Finally, one practical application is showcased where more consistent LLMs could enhance performance.

**Strengths:**

1. **Consistency Evaluation Framework**. This paper proposed an evaluation framework to quantify the consistency of LLM output. Through a comprehensive evaluation of ten LLMs and three datasets, it compares the consistency pattern across different models, the effect of CoT prompting, and transitivity as a proxy for self-agreement.

2. **Attempt to improve Consistency**. Using the win-loss rate as a re-ranking proxy, this paper demonstrates the effectiveness of aligning a more consistent LLM through data augmentation + finetuning.

3. **Good Presentation**. The logical flow of the paper is excellent and the content is well-written. I can follow the authors' points easily by linearly reading the manuscript without dragging here and there.

**Weaknesses:**

There is no significant weakness in this paper. The main contribution from my perspective is the comprehensive consistency evaluation and the derived insights. The reason I will give a 6 instead of 8 is the **novelty** contributions because of the following reasons:

1. **Reason to select these specific measurements**: In the paper the consistency metrics are straightforward but valid. The transitivity measure here is an extension of transitive ratio metrics from social network analysis and the commutativity & negation invariance are averages. I agree a good metric should be simple and direct, but the novelty contribution of the paper could be more if the authors compared different metrics and finally chose these three.

2. From Tables 2 & 3, an improved consistency could somewhat **be expected** as we have more augmented data with perfect consistency for later data types. It would be more solid to sample the same amount of perturbed data, augmented data, and augmented & negated data to make the comparison to exclude the data size as a potential factor and discuss the implications of such an experiment.

**Questions:**

1. Are transitivity, commutativity, and negation invariance relative to each other, or they are just equally important? Can we rank two LLM's consistency by just one (aggregated) aspect or we cannot easily compare two LLMs, say Phi-3-medium and Gemma-2-9B on SummEval, as their rank for transitivity and commutativity are different?

2. How to choose K in a real application? As a small K can miss cycles containing more than K items and from line 195 I understand transitivity empirically decreases as K increases, but the transitivity comparison between two LLMs can also be not unique. i.e. for K = 3 model A is better but when K = 4 model B is better.

minor: How to extract LLM's preference accurately using the prompt listed in Appendix F? If Figure 7 is the full instruction, the output should be a descriptive paragraph and I suspect both candidates's names, i.e. characters A and B, will be in the answer. Do we have detailed instructions like an output format instruction?

---

> ### Author Response · Authors · 2024-11-21
> **Response to the reviewer (1)**
>
> We sincerely thank the reviewer for their time and thoughtful feedback on our paper. We deeply appreciate the reviewer’s recognition of our work and the valuable insights they have provided. We are confident that these suggestions will enhance the quality and clarity of our paper. Below, we address each concern raised in detail. We hope that, in light of the following explanations and supplementary materials, the reviewer may consider revisiting their assessment.
>
> ---
>
> > W1: “Reason to select transitivity measurements”.
>
> We acknowledge that transitivity in Social Network Analysis (SNA) is a well-studied concept. However, **we would like to emphasize that the definition and motivation for transitivity in our framework differ significantly from those in SNA**. Below, we outline the key distinctions:
>
> - **Motivations**:
>   - **SNA Transitivity**: The SNA transitivity is used to reflect **the degree to which nodes in a graph tend to cluster together**. It is motivated by observations in real-world social networks, where there is a tendency for "friends of friends" to also be friends, leading to the formation of tightly knit groups.
>
>   - **Our Logical Transitivity**: In contrast, our logical consistency framework uses transitivity to evaluate **the degree of self-conflict in relation graphs generated by models or human judgments**. Here, transitivity is designed as a measure of logical coherence rather than clustering.
>
> - **Definition**:
>   - **SNA Transitivity**: In graph theory, transitivity quantifies the extent to which edges form closed triangles (fully connected triplets). It measures the likelihood that adjacent nodes of a given node are also connected to each other.
>
>   - **Our Logical Transitivity**: Our framework quantifies the extent to which relation graphs are acyclic. Specifically, it measures the probability that a randomly sampled set of K nodes forms acyclic relations.
>
> - **Applicability and Scenarios**:
> The differences in motivations and definitions naturally lead to distinct application contexts and methodologies:
>
>   1. **Graph Type**: SNA transitivity is primarily applied to undirected social network graphs, while our logical transitivity operates exclusively on directed relation graphs.
>
>   2. **Scope**: SNA transitivity focuses on the closure of **triplets**, whereas our logical transitivity evaluates logical circulations spanning **3 to K nodes**, with K being the size of the sampled sub-graph. *Extending SNA measures to higher-order relationships is nontrivial and computationally inefficient.*
>
> To illustrate these differences, we provide a detailed analysis of popular SNA transitivity measures:
>
>   1. Clustering Coefficient [1]: Global clustering coefficient measures
>
>      $C = \frac{\text{Number of Closed Triplets}}{\text{Number of All Triplets (closed and open)}}$
>
>      and Local clustering coefficient measures
>
>      $C_i = \frac{\text{Number of closed triplets including node i}}{\text{All triplets including node i}}$.
>
>      While applicable to directed graphs, clustering coefficient still measures the ratio between closed triplets and total triplets instead of **circular relations** and **cannot be applied to larger structures**.
>
>   2. Triad Census [2]: Counts the relative frequency of 16 possible triadic configurations in directed graphs. However, this method is inherently constrained to triads.
>
>   3. E-I (External-Internal) Index [3]: Measures the ratio of external to internal ties within a given partition, capturing group closure rather than logical acyclicity.
>
> **Novelty of Our Approach**
>
> We would like to respectfully clarify that our transitivity measure was developed independently and is not derived from existing SNA methodologies. Instead, it was **specifically designed to align with the goals of logical consistency**. As we have shown, none of the existing SNA transitivity measures can be straightforwardly extended to measure the logical transitivity of an arbitrary number of nodes.
>
> We sincerely appreciate the reviewer’s insightful suggestion, as it provides a valuable opportunity to further elucidate this distinction. To enhance the clarity of our contribution, we will include a detailed discussion of these differences in the supplementary material.
>
> ---
>
> **References**
>
> [1] Stanley Wasserman, Katherine Faust, 1994. Social Network Analysis: Methods and Applications. Cambridge: Cambridge University Press.
>
> [2] Batagelj, V., & Mrvar, A. (2001). A subquadratic triad census algorithm for large sparse networks with small maximum degree. Social networks, 23(3), 237-243.
>
> [3] Informal networks and organizational crises: An experimental simulation. David Krackhardt, Robert N. Stern - Social Psychology Quarterly, 1988, DOI:10.2307/2786835

---

> ### Author Response · Authors · 2024-11-21
> **Response to the reviewer (2)**
>
> ---
>
> > W2: “It would be more solid to sample the same amount of perturbed data, augmented data, and augmented & negated data to make the comparison to exclude the data size as a potential factor and discuss the implications of such an experiment.”
>
> We sincerely thank the reviewer for this thoughtful suggestion. We conducted the **suggested experiments** and provided the results below. Before presenting the findings, we would like to address and clarify potential misunderstandings.
>
> **Clarifications**:
>   - In Section 4, we propose a data augmentation and refinement framework to address **the challenges of obtaining high-quality, consistent data in the era of LLMs**. As acquiring such data through synthesis methods or human annotations is increasingly difficult and resource-intensive, data augmentation is a more practical and cost-effective alternative. **Our augmentation framework, which relies solely on the perturbed data, exploits its inherent information without requiring external knowledge.**
>
>   - Although the datasets differ in size, the comparisons made in Table 3 remain valid. These comparisons effectively **demonstrate the strength of our approach in utilizing and expanding the information within the perturbed data**. Nevertheless, to address the reviewer’s concerns and further evaluate the implications of data size, we conducted experiments following the suggested methodology: randomly sampling the same amount of data as the perturbed dataset, while keeping all other experimental settings consistent with those in Table 3.
>
> **Experiment Results and Observations**:
>
> The results of this experiment are presented below. We would like to emphasize the following key points:
>
>   1. **Information Expansion and Loss**:
>
>      - The augmented dataset expands the information contained in the perturbed dataset. Randomly sampling from the augmented data inherently leads to some loss of information compared to the original perturb set.
>
>      - Without employing a sophisticated sampling strategy to minimize this information loss, the comparison may not be entirely equitable.
>
>   2. **Impact on Performance**:
>      - Due to the information loss, alignment with human preferences shows a slight decrease. However, the drop in performance is relatively small.
>
>      - Despite the reduced data size, all three consistency metrics still show good improvements compared to the perturbed dataset. This demonstrates the robustness of our augmentation framework, though the performance is naturally lower than models trained on the full augmented dataset.
>
>
>
> | Models                             | Human Acc. | Stran(5) | Scomm | Sneg |
> |------------------------------------|:----------:|:--------:|:-----:|------|
> | Zero-shot inference                |    62.9    |   84.7   |  68.2 | 64.0 |
> | IT perturbed data                  |    71.3    |   92.4   |  79.6 | 60.9 |
> | IT augmented data (sampled)        |    69.1    |   96.3   |  88.9 | 61.7 |
> | IT augmented data + Neg. (sampled) |    68.8    |   96.2   |  86.2 | 84.5 |
>
>
> These findings **re-affirm the efficacy of our proposed framework** and provide further insights into its behavior under constrained data conditions. We appreciate the opportunity to conduct this analysis and will include these results in the revised version of our paper for added clarity and comprehensiveness.
>
> ---
>
> > Q1: “Are transitivity, commutativity, and negation invariance relative to each other, or they are just equally important? Can we rank two LLM's consistency by just one (aggregated) aspect or we cannot easily compare two LLMs, say Phi-3-medium and Gemma-2-9B on SummEval, as their rank for transitivity and commutativity are different?”
>
> We appreciate the reviewer’s insightful question. To clarify, we do not consider the three consistency metrics to be equally important. Their relative importance depends heavily on the specific application scenario:
>
>   - **Commutativity** becomes more critical when addressing positional bias in evaluations.
>   - **Transitivity** is essential in applications where calibration is feasible, and accurate ranking of multiple items is a priority.
>   - **Negation Invariance** is crucial in contexts requiring consistency between forward and reverse relation expressions, such as “better” versus “worse.”
>
> **We do not recommend aggregating the three metrics** into a single value, as was intentionally avoided in our paper. **Aggregation compromises their interpretability** as distinct consistency properties and offers only marginal benefits. By keeping the metrics separate, practitioners can better tailor evaluations to the specific needs of their applications.
>
> ---

---

> ### Author Response · Authors · 2024-11-21
> **Response to the reviewer (3)**
>
> ---
>
> > Q2: “How to choose K in a real application?”
>
> We thank the reviewer for this excellent question and would like to emphasize the following considerations regarding the choice of $K$.
>
>   1. **Robustness of Transitivity to $K$**:
>
>      The transitivity metric $S_{tran}(K)$ is robust to variations in $K$. It is uncommon for a model to perform well on $S_{tran}(3)$ but poorly on $S_{tran}(5)$. To illustrate this, we have added a new figure `(Fig. 9) in Appendix G`. Additionally, the figure also shows that larger values of $K$ expand the value range of $S_{tran}(K)$, making comparisons between models more distinct. For example, when $K < 5$, the $S_{tran}(K)$ of Gemma-2-9B and Phi-3-medium are very close to each other, which starts to diverge more evidently when $K >8$.
>
>   2. **Dependence on Prior Knowledge**:
>
>      Choosing K can depend on prior knowledge about the **general consistency performance of current LLMs**. For example, we observe that the consistency performance improves progressively from earlier LLMs, such as LLama-2 and Zephyr-7B-beta, to more advanced models like Gemma-2 and Llama-3. This trend suggests that considering the evaluated LLMs' expected consistency levels can inform an appropriate choice of $K$.
>
>   3. **Task-Specific Considerations**:
>
>      As discussed in Lines 301–303, consistency performance varies depending on the task. For objective tasks, consistency tends to be higher, which means task-specific nuances should also guide the selection of $K$.
>
> In summary, there is no definitive “gold standard” for choosing $K$. Similar to selecting the $N$-gram size in BLEU-$N$, **the choice depends on the models and tasks under evaluation**. We recognize the value of this discussion and have included these considerations in the `Appendix G`.
>
> ---
>
> > Minor: “How to extract LLM's preference accurately using the prompt listed in Appendix F?”
>
> We would like to thank the reviewer for observing this. We do have a format control instruction (“Please only answer with A or B.”), which is now updated in `Fig. 7`. To extract the preference, we start from the beginning of the response and try to match with char ‘A’ or ‘B’. We do notice they are tokenized differently in different LMs’ tokenizer, and we carefully specify the corresponding token for each tokenizer.
>
> We are aware that tokenization for these characters can vary across different LLM tokenizers. To address this, we carefully specify the corresponding token for each tokenizer to ensure accurate extraction. This adjustment improves the consistency and reliability of preference extraction.
>
> ---
>
> Once again, we thank the reviewer for their time and detailed feedback. We hope these clarifications address the reviewer’s concerns and welcome any additional feedback to further improve our work. Additionally, we kindly ask the reviewer to consider increasing their score in light of the explanations provided.

---

> ### Comment · Reviewer_4Hhg · 2024-11-21
>
> Thank you, authors, for the detailed response. You have addressed all of my questions thoroughly, and I’m satisfied with the results. The additional experiments and explanations are commendable and make me lean toward a score of 7. However, since ICLR only allows scores of 6 or 8, I will maintain my score at 6 for now.

---

> > ### Author Response · Authors · 2024-11-22
> > **Follow-up to reviewer's comment**
> >
> > Thank you very much for your feedback and kind words about our work. We deeply appreciate the recognition of our efforts to address your questions and provide additional experiments and explanations.
> >
> > Given your positive remarks and your indication that our revisions make you lean toward a score of 7, we kindly ask if you would consider raising the score to 8. This would be incredibly helpful for the overall evaluation of our paper, especially since a score of 6 might not fully reflect your encouraging comments.
> > Should you have any further questions or suggestions, we would be more than happy to address them.
> >
> > We truly value your time and effort in reviewing our submission and would be grateful for your consideration.
> >
> > Best,
> > Authors

---

> > > ### Author Response · Authors · 2024-12-03
> > > **Follow-Up on Review Score Reconsideration**
> > >
> > > Dear Reviewer,
> > >
> > > Thank you once again for your valuable feedback and for recognizing the merits of our work. We are pleased to see that our rebuttal has effectively addressed all of your concerns.
> > >
> > >
> > > We would like to kindly and respectfully request that you reconsider raising your score from 6 to 8, as this would significantly impact the overall evaluation of our submission. In your initial review, you highlighted concerns regarding the novelty of our approach. In our rebuttal, we carefully elaborated on the unique aspects of our method, particularly the distinction between our approach to transitivity and the traditional Social Network Analysis (SNA) transitivity. Additionally, we conducted and presented further experiments with controlled amounts of training data to strengthen our claims.
> > >
> > > We believe these efforts adequately address the novelty concerns you raised. Your acknowledgment of this progress would mean a great deal to us and reflect the substantial improvements we have made in response to your feedback.
> > >
> > >
> > > Once again, thank you for the time and effort you have dedicated to reviewing our work.
> > >
> > >
> > > Best,
> > > Authors

---

### Official Review · Reviewer_8YVz · 2024-11-03

**Soundness:** 2
**Presentation:** 3
**Contribution:** 2
**Rating:** 5
**Confidence:** 3

**Summary:**

This paper evaluates the logical consistency of LLM by preference ranking, and proposes a data-enhanced method to improve the logical consistency of LLM. Experimental results show that this method can effectively improve the logical consistency without s compromising alignment with human preferences.

**Strengths:**

* This paper expands the evaluation scenarios of logical consistency, and evaluates LLM logical consistency in three new scenarios: abstract generation evaluation, document reordering and temporal event sequence by using preference ranking method.
* A wide range of experiments are carried out, from evaluation to enhancement, to systematically explore the logical consistency of LLM

**Weaknesses:**

* In section 3.1 of the article, there is no explanation on how to use the three datasets to probe the logical consistency of LLMs. How the SummEval dataset is used to measure the negation invariance of LLMs? Is it through the form listed in Figure 1? It is suggested to supplement the relevant content.

* In section 4, the paper proposes a data refinement and augmentation method for improving the logical consistency of LLMs, which can enhance logical consistency without compromising alignment with human preferences. Human preferences are highly correlated with commutativity, and the paper does not mention that human preferences and logical consistency are contradictory relationships. In fact, human preferences should be consistent with logical consistency. Why does the paper emphasize that alignment with human preferences is not compromised?
* The related work section mentions that "most studies have concentrated on first-order relations between only two or three statements," and it seems that the paper is only exploring and improving the logical consistency of LLMs in a new application scenario, without solving the problem of "first-order relations."

**Questions:**

* In section 3.3, the article calculates the correlation between Transitivity consistency and self-agreement, and finds that they are highly correlated, and therefore concludes that “Transitivity serves as a useful proxy for evaluating the global reliability of LLMs. Is this conclusion reliable? Can Self-Agreement provide a comprehensive measure of model reliability?
* Why are the experimental parts of section 4 only conducted on SummEval? Additional experiments are suggested.

---

> ### Author Response · Authors · 2024-11-21
> **Response to the reviewer (1)**
>
> We sincerely thank the reviewer for their time and thoughtful feedback on our paper. We deeply value their insights and believe these suggestions will significantly enhance the quality and clarity of our work. Below, we address each concern raised:
>
> ---
>
> > **W1**: “In section 3.1 of the article, there is no explanation on how to use the three datasets to probe the logical consistency of LLMs. How the SummEval dataset is used to measure the negation invariance of LLMs?”
>
> We appreciate the reviewer highlighting this point. The evaluation of the three datasets is conducted using the consistency quantification methods outlined in Section 2. To clarify, we have added a detailed explanation in `Appendix F` (highlighted with red text), which explicitly describes how the consistency metrics are applied to each dataset, including measuring the LLMs’  negation invariance performance on the SummEval. We hope these revisions address the reviewer’s concerns.
>
> ---
>
> > **W2 (a)**: “In section 4, the paper proposes a data refinement and augmentation method for improving the logical consistency of LLMs, which can enhance logical consistency without compromising alignment with human preferences. Human preferences are highly correlated with commutativity, and the paper does not mention that human preferences and logical consistency are contradictory relationships. In fact, human preferences should be consistent with logical consistency. Why does the paper emphasize that alignment with human preferences is not compromised?”
>
> **We respectfully request further clarification regarding this concern,** as there might be potential misunderstanding of our paper. Our paper does not claim or suggest that “human preferences and logical consistency are contradictory relationships.” In fact, we do not believe this statement is well-supported. Additionally, the reviewer’s assertion that “human preferences should be consistent with logical consistency” appears self-contradictory and is also not stated in our paper. We would appreciate further elaboration on the specific questions raised here.
>
> ---
>
> > **W2 (b)**: “Why does the paper emphasize that alignment with human preferences is not compromised?”
>
> This statement, found in lines 530-533 [1], specifically refers to the experimental results in Section 4, as presented in Table 3. We emphasize that “human preferences are not compromised” because the results demonstrate that **our data augmentation framework improves logical consistency while maintaining the same level of accuracy in human preference alignment compared to models trained on perturbed data**. We believe this comparison is both fair and well-supported by the data presented.
>
> ---
>
> > **W3 (a)**: the paper is only exploring and improving the logical consistency of LLMs in a new application scenario.
>
> We respectfully disagree with this observation. Our evaluation framework is **not limited to a single application scenario but is broadly applicable to any task where logical relations are relevant**.
>   - For instance, it can be utilized in scenarios involving preference rankings of response candidates, relevance judgments over retrieved documents, temporal or causal relationships among events, hierarchical relationships among entities, or entailment relationships between statements, as discussed in Lines 111–116. This generality is one of the key advantages of our framework, compared with previous works.
>
> ---
>
> > **W3 (b)**: Concerns about first order relations. “it seems that the paper is only exploring and improving the logical consistency of LLMs in a new application scenario, without solving the problem of "first-order relations."
>
> We would like to clarify our use of the term “first-order relations.” In our paper, this refers to the logical relationships between two or three statements, such as the transitivity of a triplet. Unlike previous works, which are mostly constrained to first-order logical relationships, our framework addresses **the logical consistency across an arbitrary number of statements or items**. This capability is novel and represents a significant step forward.
>   - To avoid potential ambiguity, we have revised the corresponding paragraph (Line 517) to explicitly define “first-order relations.”
>
> ---
> ---
>
>  **Reference**
>
>   - [1] Line 530-533: “To improve the logical consistency of LLMs, we then proposed a data refinement and augmentation framework, which effectively reduces logical inconsistencies at the source: directly in the data. Our experimental results showed that models trained on this refined and augmented data achieve improved logical consistency without compromising alignment with human preferences.”

---

> ### Author Response · Authors · 2024-11-21
> **Response to the reviewer (2)**
>
> ---
>
> > **Q1**: In section 3.3, the article calculates the correlation between Transitivity consistency and self-agreement, and finds that they are highly correlated, and therefore concludes that “Transitivity serves as a useful proxy for evaluating the global reliability of LLMs. Is this conclusion reliable? Can Self-Agreement provide a comprehensive measure of model reliability?
>
> We agree with the reviewer’s point that our initial conclusion may overstate the relationship between self-agreement and global reliability. Upon reflection, we have revised the term **“global reliability”** to **“local robustness”** to more accurately represent our findings (Line 322). We believe this adjustment better aligns the conclusion with the scope of our experimental evidence.
>
> ---
>
> > **Q2**: “Why are the experimental parts of section 4 only conducted on SummEval?”
>
> **We would like to clarify a misunderstanding here.** In Section 4, all experiments are conducted on the *Summary with Feedback* dataset (as shown in Table 3 and Section 4.2), not SummEval. We chose this dataset because of its large size and the breadth of topics it covers, which we believe provide sufficient diversity for evaluating our methods.
>
> However, to further address the reviewer’s concerns, we have conducted **additional experiments** using a different dataset, *MS-MARCO* (details explained in Appendix B), with all other experimental setups remaining consistent with Table 3 in the manuscript. (Also perturb 10% of the pairwise comparison labels.)
>
> | Models                   | Human Acc. | $S_{tran}(5)$ | $S_{comm}$ | $S_{neg}$ |
> |--------------------------|:----------:|:--------:|:-----:|------|
> | Zero-shot inference      |    57.2    |   74.2   |  68.5 | 64.7 |
> | IT perturbed data        |    74.7    |   86.7   |  81.1 | 62.9 |
> | IT augmented data        |    75.0    |   91.3   |  85.9 | 63.1 |
> | IT augmented data + Neg. |    74.9    |   90.2   |  86.2 | 85.5 |
>
> The results from these experiments **confirm the robustness of our findings with the results on summary with feedback**, where our data augmentation framework can successfully improve the logical consistency while not loosing alignment with human preference. This experiment reinforces the generalizability of our framework.
>
> ---
> ---
>
> Once again, we thank the reviewer for their time and detailed feedback. We hope these clarifications address the reviewer’s concerns and welcome any additional feedback to further improve our work. We will further revise our manuscript according to the discussion. Additionally, we kindly ask the reviewer to consider revising their score in light of the explanations provided.

---

> ### Author Response · Authors · 2024-11-27
> **Reminder to the Reviewer**
>
> As the discussion period nears its conclusion, we would like to kindly remind you to review our rebuttal and share your thoughts. **We would like to confirm whether our responses adequately address your concerns.** If there are remaining issues with the paper, we are open to further clarifications. Otherwise, **we respectfully request the reviewer to consider revisiting the score in light of our responses**.
>
> ---
>
> To further strengthen the paper, we have conducted additional experiments, which we believe may also be of interest:
>
> 1. Beyond the above experiments on *MS-MARCO*, we demonstrate that models trained with augmented *MS-MARCO* data achieve improved performance using the PairS algorithm as well (aligned with Table 4 in the paper).
>
> | Model             | Human | $S_{tran}(5)$ | $S_{comm}$ | PairS | PairS calibrated |
> |-------------------|:-----:|:--------:|:-----:|:-----:|:----------------:|
> | zero-shot         |  57.2 |   74.2   |  68.5 |  18.7 |       22.1  (+3.4)     |
> | IT perturbed data |  74.7 |   86.7   |  81.1 |  58.1 |       60.3  (+2.2)     |
> | IT augmented data |  75.0 |   91.3   |  85.9 |  61.0 |       62.6   (+1.6)    |
>
>
> 2. We have also explored alternative ranking estimation methods, including the ELO rating system and the Bradley-Terry model, comparing their performance against the Win-loss rate.
>
>
> | Models              | Human Acc. | Stran(5) | Scomm |
> |---------------------|:----------:|:--------:|:-----:|
> | Zero-shot inference |    62.9    |   84.7   |  68.2 |
> | IT perturbed data   |    71.3    |   92.4   |  79.6 |
> | IT augmented **W-L**    |    70.5    |   97.5   |  89.4 |
> | IT augmented **ELO**    |    71.4    |   97.7   |  89.7 |
> | IT augmented **B-T**    |    71.2    |   98.6   |  91.0 |
>
> These additional results **consistently support our conclusions and further validate the effectiveness of our data augmentation framework**.
>
> ---
>
> We sincerely appreciate the time and effort you have devoted to reviewing our work and look forward to your feedback.
>
> Best regards,
>
> Authors

---

> > ### Author Response · Authors · 2024-12-02
> > **Reminder to the Reviewer**
> >
> > As the discussion period comes to a close, we kindly remind you to review our rebuttal and share your feedback. We would appreciate confirmation on whether our responses sufficiently address your concerns. If there are any remaining issues with the paper, we are happy to provide further clarifications. Otherwise, we respectfully request that you consider revisiting the score based on our responses.

---

### Official Review · Reviewer_prZw · 2024-11-04

**Soundness:** 3
**Presentation:** 4
**Contribution:** 4
**Rating:** 8
**Confidence:** 3

**Summary:**

This paper's main achievement is the introduction of a method to measure the degree of consistency of a LMM, and a method to improve it. The author(s) starts by identifying three ways an LMM might not be consistent, and proposes ways to numerically evaluate the degree to which each of these inconsistencies might arise. They then go on to propose a way to improve an LLM's consistency by data refinement and augmentation,

**Strengths:**

In my opinion this is a good paper.
The paper is very clearly written. The author(s) succeeded very well in conveying the main ideas. They present their arguments with clear figures.
I also appreciate the definitions of s_tran, s_comm and s_neg, which make great intuitive sense to me. The authors provide clear tables that summarise their empirical findings.
The author(s) makes very clear what the objective of their paper is, and they clearly reach this goal in a very satisfying way. They motivate their approach well. The given empirical data support the claims well. Given the current interest in LLMs, I believe this work is significant for the ICLR community.

**Weaknesses:**

I here list the weaknesses I see in this paper. Note that I believe that they are less important than the strengths.

In the introduction, line 048, the author(s) says that consistency of its predictions is the foundation of a reliable and trustworthy system. While I certainly agree that this is *one of* the foundations, I don't think consistency alone is *the* foundation. What about a very consistently wrong system? One that consistently predicts falsehood? Certainly it is internally consitent, but it fails to accurately describe the world. I would say that calibration (or truth tracking, or something along those lines) is equally foundational to reliability and trustworthiness. If the author(s) agrees, I suggest to change this accordingly.
What the authors mention on line 087 seems relevant for this, too.

On Page 3, line 150, the autor(s) mentions that if F is transitive, the corresponding relation graph is a DAG. I think this is not entirely accurate: it is a DAG when F is transitive *and irreflexive*. The author(s) seems to tacitly assume that F is irreflexive, but they should at least mention this somewhere. Also, why do they assume that this cannot be a way to measure internal consistency?
A similar remark holds in Fig 2 (b) and (d): in normal parlance, the red cycle is not intranstive, but it's just not a cycle of an order that is both transitive and irreflexive.
I wonder if the authors should not clarify this. However, even if they decide not to, despite my disagreement here, I do believe that the measure s_tran makes very good sense.

On Page 4, line 191, I think that the wording is slightly too strong, in two different ways: the metric s_tran(K) being 1 does not represent perfect transitivity, but rather (i) very likely (ii) transitivity *of cycles of size K*.

In Eqs (2) and (3), the denominator should both read |X| (|X|-1) instead of |x|(|x|-1) (so with a capital X.

On line 228, there is a space missing between "relation" and "(e.g., reversing".

One line 304, the author(s) talks about CoT prompting, which I believe stands for Chain of Thought prompting. However, this is nowhere introduced in the paper. I suggest that they introduce this somewhere.

**Questions:**

My only question concerns the robustness of the wording of the prompting to the results. When I use an LLM, I feel that the results, including the consistency, depends heavily on my wording. Does the author(s) have a way to measure this robustness? Are there results about this? This question naturally comes to mind when reading Sections 3 and 4, so having some ideas about this might improve the paper. Naturally, I would completely understand if this falls outside the scope of the current work.

---

> ### Author Response · Authors · 2024-11-24
> **Response to the reviewer (1)**
>
> We sincerely thank the reviewer for their time and effort in evaluating our work. We greatly appreciate the positive recognition of our contributions, as well as the insightful feedback provided. We believe these suggestions will significantly enhance the quality and clarity of our paper. Below, we address each concern in detail:
>
> ---
>
> > W1: “In the introduction, line 048, the author(s) says that consistency of its predictions is the foundation of a reliable and trustworthy system. While I certainly agree that this is one of the foundations, I don't think consistency alone is the foundation. What about a very consistently wrong system? One that consistently predicts falsehood? Certainly it is internally consistent, but it fails to accurately describe the world. I would say that calibration (or truth tracking, or something along those lines) is equally foundational to reliability and trustworthiness. If the author(s) agrees, I suggest to change this accordingly. What the authors mention on line 087 seems relevant for this, too.”
>
> We fully agree with the reviewer’s observation that both consistency and accuracy are critical for the reliability and trustworthiness of a system. As noted in the footnote on line 053, we state that “In the fields of behavioral economics and psychology, systems are traditionally evaluated on two key dimensions: validity and reliability.” We appreciate the reviewer highlighting this point and will revise the statement in line 048 to more accurately reflect our intended meaning by emphasizing the equal importance of both consistency and accuracy as foundational dimensions.
>
> ---
>
> > W2: On Page 3, line 150, the autor(s) mentions that if F is transitive, the corresponding relation graph is a DAG. I think this is not entirely accurate: it is a DAG when F is transitive and irreflexive. The author(s) seems to tacitly assume that F is irreflexive, but they should at least mention this somewhere. Also, why do they assume that this cannot be a way to measure internal consistency?
>
> We thank the reviewer for this insightful observation. You are correct that our statement assumes F is irreflexive, although this was not explicitly stated in the paper. For instance, in Figure 3 and Equations 2 and 3, we omit the diagonal elements of the pairwise comparison matrices, and we compute the total pairwise comparisons as $|X| \cdot (|X|-1)$ instead of $|X|^2$. We will revise the manuscript to explicitly clarify this assumption and provide the rationale behind it.
>
> Regarding the suggestion that reflexive consistency could serve as a measure of internal logical consistency, we agree in principle. However, in practical scenarios—such as evaluating causal or temporal orders, or comparing preferences—it is uncommon to compare an item against itself when working with large language models (LLMs). As a result, our evaluation framework assumes F is irreflexive. That said, we appreciate this important point and will include a discussion of reflexive consistency, along with our motivations for adopting the irreflexive assumption.
>
> ---
>
> > Others: Line 191, line 228, line 304 and Eqs 2 &3.
>
> We appreciate the reviewer’s careful corrections, and we have revised the paper accordingly.
>
> ---
>
> > Q1: Concerns the robustness of the wording of the prompting to the results.
>
> We appreciate the reviewer’s insightful observations. It is indeed true that LLMs exhibit varying levels of sensitivity to prompt phrasing. As illustrated in Figure 5, where each dot in the subplots represents a semantically equivalent but differently phrased prompt, variations in prompt wording can influence LLMs’ accuracy relative to human judgments and their commutativity performance. **The results presented in Table 1 are based on a single, carefully crafted prompt instruction (shown in Figure 7) that is designed to be broadly interpretable by most LLMs**.
>
> Interestingly, we observed that **transitivity values are less affected by prompt variations compared to commutativity**. To illustrate this, we computed the range of transitivity values using the same 10 prompt variations from the experiment depicted in Figure 5. The findings show that transitivity is relatively more robust to prompt variations, and more irrelevant to human preference accuracy.
>
> | Model        | SummEval  | NovelEval | CaTeRS    |
> |--------------|-----------|-----------|-----------|
> | Llama-2-7B   | 86.4+-3.4 | 70.1+-4.3 | 87.0+-2.1 |
> | Llama-3-8B   | 92.1+-3.3 | 80.6+-6.7 | 90.9+-3.0 |
> | Mistral-7B   | 94.6+-2.5 | 92.1+-3.4 | 95.3+-2.8 |
> | Phi-3-medium | 96.0+—0.9 | 94.1+-0.8 | 94.4+-0.5 |
> | Gemma-2-9B   | 95.6+-1.2 | 92.2+-1.5 | 95.9+-0.9 |
>
> ---

---

> ### Author Response · Authors · 2024-11-24
> **Response to the Reviewer (2)**
>
> > Q2: Is there a way to measure the prompt sensitivity.
>
> We agree with the reviewer that prompt sensitivity is a **critical dimension of LLM robustness**. While our primary focus in this work is more on **logical consistency**, we acknowledge that prompt sensitivity can intersect with this topic and is an important consideration. Although we did not explicitly measure or analyze prompt sensitivity as part of our study, Figure 5 indirectly addresses this issue by showing performance variations across different prompt formulations.
>
> We note that several existing works have specifically explored methods for quantifying prompt sensitivity, which may offer complementary insights into this area. In future work, incorporating a more explicit investigation of prompt sensitivity could provide a valuable extension to our current analysis.
>
> ---
>
> Once again, we sincerely thank the reviewer for their thoughtful feedback. Your suggestions have provided us with valuable perspectives to enhance the robustness and clarity of our work.

---

> > ### Comment · Reviewer_prZw · 2024-11-25
> > **Response to the authors**
> >
> > I'd like to thank the authors for their very detailed replies to my comments. I'm completely satisfied with their replies, and hence I remain of the opinion that this is a good paper.

---

### Meta-Review · Area_Chair_JN6D · 2024-12-20

**Metareview:**

This paper considers measuring logical consistency of LLMs using three metrics, transitivity, commutativity, and negation invariance, and proposed a method for improving LLM consistency through data refinement and augmentation. There are diverse opinions from reviewers, and this paper was rather borderline. The problem of interest is important for LLMs, the considered metrics verify that the current LLMs don't show consistent behavior, and the empirical performance of the proposed methods is promising in improving the consistency. While certain issues remain. Both choice and originality of the considered three metrics have been questioned by reviewers. It is not unclear if the considered three metrics are sufficient for LLM's consistency and the main conceptual contribution over the existing literature. The proposed method in terms of refinement and augmentation is not very exciting. Moreover, the extension and generality of the proposed method to broader cases is not straightforward, and the improvement space of presentation exists. I think the paper does have an interesting contribution and encourage the authors to take into consideration all the feedback provided by the reviewers to strengthen their manuscript for resubmission.

**Additional Comments On Reviewer Discussion:**

Though part of reviewers' concerns have been nicely resolved during rebuttal and some reviewers are satisfied with the rebuttal, the main concerns from reviewers remain, including the choice and originality of the considered three metrics, the main conceptual/methodology contribution over the existing literature, the extension and generality of the proposed method to broader cases, and the presentation. Considering these factors, this paper is not quite ready for publication.

---

### Decision · Program_Chairs · 2025-01-22

Reject